# Making Sense of Dependence: Efficient Black-box Explanations Using Dependence Measure

**Paul Novello** [1] [2]      **Thomas Fel** [2] [3]      **David Vigouroux** [1] [2]

[1] IRT Saint Exupery, France, [2] Artificial and Natural Intelligence Toulouse Institute,
Université de Toulouse, France [3] Carney Institute for Brain Science, Brown University, USA
`paul.novello@irt-saintexupery.com`

## Abstract

This paper presents a new efficient black-box attribution method built on Hilbert-Schmidt Independence Criterion (HSIC). Based on Reproducing Kernel Hilbert Spaces (RKHS), HSIC measures the dependence between regions of an input image and the output of a model using the kernel embedding of their distributions. It thus provides explanations enriched by RKHS representation capabilities. HSIC can be estimated very efficiently, significantly reducing the computational cost compared to other black-box attribution methods. Our experiments show that HSIC is up to 8 times faster than the previous best black-box attribution methods while being as faithful. Indeed, we improve or match the state-of-the-art of both black-box and white-box attribution methods for several fidelity metrics on Imagenet with various recent model architectures. Importantly, we show that these advances can be transposed to efficiently and faithfully explain object detection models such as YOLOv4. Finally, we extend the traditional attribution methods by proposing a new kernel enabling an ANOVA-like orthogonal decomposition of importance scores based on HSIC, allowing us to evaluate not only the importance of each image patch but also the importance of their pairwise interactions. Our implementation is available at `https://github.com/paulnovello/HSIC-Attribution-Method`.

## 1 Introduction

Artificial Intelligence has established itself as the reference technique for tackling many real-world automation tasks. Consequently, the diversity of its applications is growing and reaching fields where its outputs can contribute to critical decision-making. In such cases, it is essential to be able to provide explanations for each link of the decision chain, including AI algorithms. Over the past decade, many techniques have emerged to explain the predictions of these algorithms [43, 36, 41, 15, 52, 32, 31, 27, 26, 38], marking the birth of a new field called Explainable Artificial Intelligence (XAI). The tools developed in this research field, mostly designed to explain neural networks, have already proven helpful. For instance, it has been used in model debugging, identification of new development strategies for practitioners, and failure understanding.

Initial approaches are based on analyzing the internal state of neural networks during inference, often relying on input gradients or activation values of hidden layers [43, 41, 15, 25]. However, the gradient only reflects the model's operation in an infinitesimal neighborhood around an input and can therefore be misleading [18]. Furthermore, their applicability is limited to the case where the final user has access to the implementation of the model. Therefore, such methods cannot be applied in the most common use cases, e.g. when models are made available by third parties through API calls or specialized hardware. In order to address this issue, some black-box approaches have

36th Conference on Neural Information Processing Systems (NeurIPS 2022).

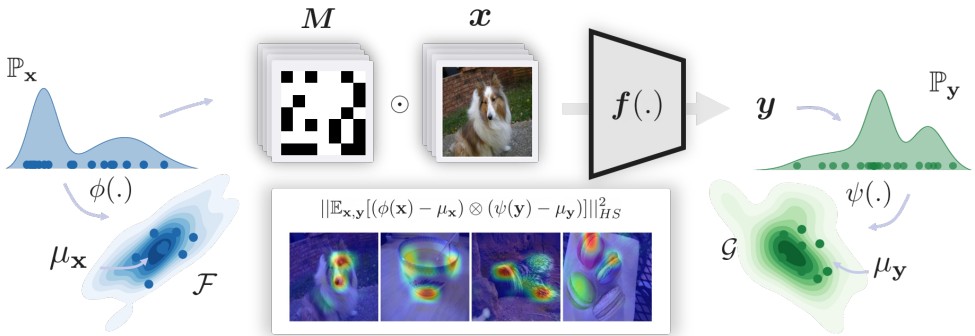

Figure 1: **HSIC Explainability method**. We sample random binary masks $\mathbf{M}$ that we use to perturb the input image $\boldsymbol{X}$. We obtain a perturbed output and measure the dependence between the distribution of each patch $\mathbb{P}_M$ of the binary mask and that of the output $\mathbb{P}_\mathbf{y}$. We use a dependence measure, Hilbert Schmidt Independence Criterion (HSIC), based on the kernel embedding of this distribution in a Reproducing Kernel Hilbert Space (RKHS). Each patch is then assigned the value of this measure: the more independent a patch is from $\mathbf{y}$, the less important it is to explain it.

been recently proposed, relying on the perturbation of the input and the observation of its effect on the output [55, 36, 32, 11]. One challenge of such perturbation methods is assessing this effect together with taking into account complex interactions inherent to deep neural networks. To account for these characteristics, black-box methods resort to complex Monte Carlo methods that require a high number of model forward passes, which can be expensive for recent neural networks that are growing larger.

In [11, 29], the authors propose methods to reduce the required number of forward passes, but the obtained performance improvements still do not make them close to white-box methods. In this work, similarly to [11], we cast perturbation studies as Global Sensitivity Analysis (GSA) [34]. However, we rely on a whole different approach based on dependence measure rather than analysis of variance. We measure the dependence between patch-wise perturbations of an input image and the model's output by comparing the distribution of perturbed inputs and outputs embedded in a Reproducing Kernel Hilbert Space (RKHS). More specifically, we use Hilbert-Schmidt Independence Criterion (HSIC), a dependence measure based on the Hilbert-Schmidt norm of the empirical cross-covariance operator evaluated between the represented distribution. HSIC leverages the rich theory of RKHS, thereby capturing more diverse information than variance-based indices such as Sobol. In addition, it can be estimated more efficiently, even bridging the performance gap between black-box and white-box methods.

Our contributions are as follows: **(1)** we introduce a new efficient black-box attribution method relying on HSIC; **(2)** we derive a new kernel that confers an Analysis of Variance (ANOVA)-like orthogonal decomposition property, allowing us to go beyond usual attribution methods and evaluate interactions between patches of the image; **(3)** we conduct experiments to assess the fidelity of our method on ImageNet and show that it improves or matches the state-of-the art for different metrics while bridging the computational gap between black-box and white-box attribution methods; **(4)** we demonstrate its versatility and its potential by successfully applying it to a less common test case: explanations of object detection; and a new test case: evaluation of pairwise interactions between patches of the input image.

## 2 Related work

Our work builds on prior efforts aiming to develop attribution methods in order to explain the prediction of a deep neural network by pointing to input variables that support the prediction – typically pixels or groups of pixels, i.e. patches in the image – which lead to importance maps.

**Attribution methods for white-box models** A large number of attribution methods have been developed relying on the gradient of the decision studied. The first method was introduced in [3] and improved in [43, 55, 50] and consists of explaining the decisions of a convolution model by

back-propagating the gradient from the output to the input, indicating which pixels affect the decision score the most. However, this family of methods is limited because they focus on the influence of individual pixels in an infinitesimal neighborhood around the input image in the image space. For instance, it has been shown that gradients often vanish when the prediction score to be explained is near the maximum value [52]. Integrated Gradient [52] and SmoothGrad [45] partially address this issue by accumulating gradients. Another family of attribution methods relies on the neural network's activations. Popular examples include CAM [56], which computes an attribution score based on a weighted sum of feature channel activities – right before the classification layer. GradCAM [41] extends CAM via the use of gradients, reweighting each feature channel to take into account their importance for the predicted class. Nevertheless, the choice of the layer has a huge impact on the quality of the explanation. In comparison, our proposed approach is model-agnostic and hence does not require access to internal computations.

**Attribution methods for black-box models**   In this paper, we extend the problem by restricting it to a black-box model: the analytical form and potential internal states of the model are unknown. Several methods compute influence scores for each individual pixel or group of pixels.

The first method, Occlusion [55], masks individual image regions – one at a time – with an occluding mask set to a baseline value and assigns the corresponding prediction scores to all pixels within the occluded region. Then the explanation is given by these prediction scores and can be easily interpreted. However, occlusion fails to account for the joint (higher-order) interactions between multiple image regions. For instance, occluding two image regions – one at a time – may only decrease the model's prediction minimally (say a single eye or mouth component on a face), while occluding these two regions together may yield a substantial change in the model's prediction if these two regions interact non-linearly as is expected for a deep neural network. Our work, together with related methods such as LIME [36], RISE [32] and more recently Sobol [11] addresses this problem by randomly perturbating the input image in multiple regions at a time.

Surprisingly, RISE [32] and Sobol Attribution [11] have recently shown that black-box attribution methods can rival and even surpass the white-box methods commonly used without recourse to internal states. However, despite the efforts in [11] to limit their computational overhead, black-box methods remain far from white-box methods in terms of execution time. In this work, we show that it is possible to match or even surpass the performances of current black-box methods while reaching computation times lower than some white-box methods by using dependence measure-based Global Sensitivity Analysis (GSA).

**Global sensitivity analysis using dependence measures**   Our attribution method builds on the GSA framework. The approach was introduced in the 70s [7] and was popularized with variance-based sensitivity analysis and Sobol indices [46]. It consists of evaluating the sensitivity of a model's output of interest to some input design variables. GSA is currently used in many fields, especially for the study of physical phenomena [24, 34]. Recently, dependence measure-based sensitivity analysis was introduced in [8] and was shown to circumvent some practical issues of variance-based sensitivity analysis. In particular, by relying on the representation capabilities of RKHS, the dependence measure that we use in this work, HSIC [20], captures more diverse information than traditional variance-based indices for far fewer model evaluations.

## 3   Explanations using Hilbert-Schmidt Independence Criterion

In this section, we describe sensitivity analysis-based attribution methods, define Hilbert-Schmidt Independence Criterion (HSIC) [20] and explain how we can use it and adapt it to design a new black-box attribution method whose efficiency competes with white-box methods. We also explain the theoretical advantages of HSIC that we can build on to go beyond traditional attribution methods.

### 3.1   Sensitivity analysis of perturbed black-box models

Let $\boldsymbol{f} : \mathcal{X} = \mathcal{X}_1, ..., \mathcal{X}_n \to \mathcal{Y}$ be the model under study, $x_i \in \mathcal{X}_i$ the input variables and $\boldsymbol{y} = \boldsymbol{f}(x_1, ..., x_n) \in \mathcal{Y}$ the output value of the model $\boldsymbol{f}$. GSA studies the sensitivity of $\boldsymbol{y}$ to each input $x_i$ by considering them as iid (independent and identically distributed) random variables and assessing the link between their distribution and that of the output after an initial input sampling. Given an input

vector $\boldsymbol{X} = (x_1, ..., x_n)$, a prediction $\boldsymbol{y} = \boldsymbol{f}(\boldsymbol{X})$ can thus be explained using sensitivity analysis by applying random perturbations $\mathbf{x} = (\mathrm{x}_1, ..., \mathrm{x}_n), \mathrm{x}_i \sim \mathbb{P}_{\mathcal{X}_i}$ of the original $\boldsymbol{X}$ and analyzing the importance of each $\mathrm{x}_i$ for explaining the variations of $\boldsymbol{y}$ - which is considered a random variable, $\mathbf{y} \sim \mathbb{P}_{\mathbf{y}}$.

For image data, the inputs $x_i$ are pixels. However, pixel perturbations would only emphasize low level explanations. To obtain high level and meaningful explanations, we rather consider a random perturbation mask $\boldsymbol{M} = (M_1, ..., M_d) \in [0, 1]^d$. We upsample this mask using a Nearest Neighbor interpolation method to obtain $u(\boldsymbol{M}) \in [0, 1]^n$, a patch-perturbated vector that we apply on the input image $\boldsymbol{X}$ using a mask operator $\boldsymbol{\tau} : \mathcal{X} \times [0, 1]^d \to \mathcal{X}$. More specifically, we use the inpainting operator defined by $\boldsymbol{\tau}(\boldsymbol{X}, \boldsymbol{M}) = \boldsymbol{X} \odot u(\boldsymbol{M}) + (1 - u(\boldsymbol{M}))\mu$, with $\odot$ the Hadamard product and $\mu$ a baseline value (here, $\mu$ is a black image with all pixels' value $= 0$ [36, 55]). Hence, the mask $\mathbf{M}$ aggregates the patch-wise random perturbations $M_i$ that are sampled independently for each patch ($M_i$ are iid). In practice, the perturbations contained in the mask are binary perturbations, to simulate whether the information contained in the patch is kept in the image or not.. We thereby assess the effect of each image patch, represented by $M_i$, on the output.

The perturbation methodology thus consists of **(1)** sampling $p$ masks $\{\boldsymbol{M}^{(1)}, ..., \boldsymbol{M}^{(p)}\}$ from $\mathbf{M} \sim \mathbb{P}_{\mathbf{M}}$ (with $\mathbb{P}_{\mathbf{M}} = \mathbb{P}_{M_1} \times ... \times \mathbb{P}_{M_p}$), **(2)** applying them to the original input vector, leading to $p$ perturbed input vectors (e.g., partially masked images) $\{\boldsymbol{\tau}(\boldsymbol{X}, \boldsymbol{M}^{(1)}), ..., \boldsymbol{\tau}(\boldsymbol{X}, \boldsymbol{M}^{(p)})\}$ **(3)** computing the predictions $\{\boldsymbol{y}^{(1)}, ..., \boldsymbol{y}^{(p)}\} = \{\boldsymbol{f}(\boldsymbol{\tau}(\boldsymbol{X}, \boldsymbol{M}^{(1)})), ..., \boldsymbol{f}(\boldsymbol{\tau}(\boldsymbol{X}, \boldsymbol{M}^{(p)}))\}$ and **(4)** statistically assessing the effect of each mask $M_i$ on $\boldsymbol{y}$ by estimating a sensitivity measure between each $\mathbb{P}_{M_i}$ and $\mathbb{P}_{\mathbf{y}}$ from the previous sampling. In this paper, we consider that the more independent $M_i$ is from $\mathbf{y}$, the less important the corresponding image patch is to explain it [8]. In the following, we describe HSIC, a dependence measure, and how to use it in practice.

## 3.2 Hilbert-Schmidt Independence Criterion

Let $\mathbf{x}$ and $\mathbf{y}$ be two random variables of probability distribution $\mathbb{P}_{\mathbf{x}}$ and $\mathbb{P}_{\mathbf{y}}$ defined on $\mathcal{X}$ and $\mathcal{Y}$. HSIC measures the dependence between $\mathbb{P}_{\mathbf{x}}$ and $\mathbb{P}_{\mathbf{y}}$ based on their embedding in Reproducing Kernel Hilbert Space (RKHS). Let $\varphi : \mathcal{X} \to \mathcal{F}$ and $\psi : \mathcal{Y} \to \mathcal{G}$ two continuous feature mapping between $\mathcal{X}$, $\mathcal{Y}$, and two RKHS $\mathcal{F}, \mathcal{G}$, such that the inner product between the feature embeddings of $x, x' \in \mathcal{X}$ in $\mathcal{F}$ is given by the kernel $k(x, x') = \langle \varphi(x), \varphi(x') \rangle$ (and $l(y, y') = \langle \psi(y), \psi(y') \rangle$ for $y, y' \in \mathcal{Y}$). The cross-covariance operator $C_{\mathbf{xy}} : \mathcal{G} \to \mathcal{F}$ between the random variables $\mathbf{x}$ and $\mathbf{y}$ is defined in [17] and can be written.

$$C_{\mathbf{xy}} = \mathbb{E}_{\mathbf{xy}}[(\varphi(x) - \mu_{\mathbf{x}}) \otimes (\psi(y) - \mu_{\mathbf{y}})],$$

where $\mu_{\mathbf{x}} = \mathbb{E}_{\mathbf{x}}[\varphi(x)]$ and $\mu_{\mathbf{y}} = \mathbb{E}_{\mathbf{y}}[\psi(y)]$ are the mean embedding of $\mathbf{x}$ and $\mathbf{y}$ in $\mathcal{F}$ and $\mathcal{G}$. When $\mathcal{F}$ and $\mathcal{G}$ are universal RKHS on the compact domains $\mathcal{X}$ and $\mathcal{Y}$, then $\|C_{\mathbf{xy}}\|_{HS} = 0$ if and only if $\mathbf{x}$ and $\mathbf{y}$ are independent, where $\| \cdot \|_{HS}$ denotes the Hilbert Schmidt norm (see [21]). In [20], the authors define HSIC as $\|C_{\mathbf{xy}}\|_{HS}^2$, which can be written:

$$
\begin{aligned}
HSIC(\mathbf{x}, \mathbf{y}) = & \mathbb{E}_{\mathbf{x}, \mathbf{x}', \mathbf{y}, \mathbf{y}'}[k(x, x')l(y, y')] + \mathbb{E}_{\mathbf{x}, \mathbf{x}'}[k(x, x')]\mathbb{E}_{\mathbf{y}, \mathbf{y}'}[l(y, y')] \\
& - 2\mathbb{E}_{\mathbf{x}, \mathbf{y}}[\mathbb{E}_{\mathbf{x}'}[k(x, x')]\mathbb{E}_{\mathbf{y}'}[l(y, y')]],
\end{aligned}
\tag{1}
$$

where $\mathbf{x}, \mathbf{x}'$ and $\mathbf{y}, \mathbf{y}'$ are pairwise iid. HSIC can also be expressed in terms of Maximum Mean Discrepancy (MMD), which is a distance between mean embeddings defined in an RKHS [51]. More specifically, let the product RKHS $\mathcal{P} = \mathcal{F} \times \mathcal{G}$ with kernel $v((x, y), (x', y')) = k(x, x')l(y, y')$. Then, $HSIC(\mathbf{x}, \mathbf{y}) = \gamma_v^2(\mathbb{P}_{\mathbf{x}}\mathbb{P}_{\mathbf{y}}, \mathbb{P}_{\mathbf{x}, \mathbf{y}})$, where $\gamma_v$ is the MMD operator on $\mathcal{P}$. HSIC thus measures the distance between $\mathbb{P}_{\mathbf{xy}}$ and $\mathbb{P}_{\mathbf{y}}\mathbb{P}_{\mathbf{x}}$ embedded in $\mathcal{P}$ [8]. Since $\mathbf{x} \perp \mathbf{y} \Rightarrow \mathbb{P}_{\mathbf{xy}} = \mathbb{P}_{\mathbf{y}}\mathbb{P}_{\mathbf{x}}$, the closer these distributions are, in the sense of the MMD, the more independent they are.

Thus, given a set of inputs $\{\boldsymbol{x}_1, ..., \boldsymbol{x}_p\}$ and the associated outputs $\{\boldsymbol{y}_1, ..., \boldsymbol{y}_p\}$, [20] shows that HSIC can be estimated by

$$\mathcal{H}_{\mathbf{x}, \mathbf{y}}^p = \frac{1}{(p-1)^2} \operatorname{tr}(KHLH), \tag{2}$$

where $H, L, K \in \mathbb{R}^{p \times p}$ and $K_{ij} = k(x_i, x_j), L_{i,j} = l(y_i, y_j)$ and $H_{ij} = \delta(i = j) - p^{-1}$. Using this formula, $\mathcal{H}_{\mathbf{x}, \mathbf{y}}^p$ can be computed with a $\mathcal{O}(p^2)$ complexity. In this work, the input variables $M_i$ are the patch perturbations. Therefore, we compute $\mathcal{H}_{M_i, \mathbf{y}}^p$ i.e. the estimation of the HSIC between a patch $M_i$ and the output $\mathbf{y}$, for each patch ($i \in \{1, ..., d\}$), see Algorithm 1. We denote

$\mathcal{H}_i^p := \mathcal{H}_{M_i,\mathbf{y}}^p$ for clarity. In the next section, we discuss the kernels $k$ and $l$ and show that we can obtain a valuable ANOVA-like orthogonal decomposition property for HSIC-based indices, allowing to assess interactions between input variables.

### 3.3 Orthogonalisation of HSIC to enable evaluation of interactions

One question of interest in explainability is the measurement of the importance of a specific group of variables. Indeed, it is notorious that neural networks are highly non-linear, and as it has been demonstrated in several works [14, 11], the effects of the groups of variables are far from being additive. Concretely, some areas of the image may only be important in interaction with other areas, affecting the output only when both areas are perturbed at the same time - as we shall see in Section 4.4 (for instance, for mustaches of a puma).

Let $\{x_1, ..., x_n\} \in \mathcal{X}^n$ be a set of $n$ univariate random variables. For any subset $A = \{l_1, ..., l_{|A|}\} \subseteq \{1, ..., n\}$, we denote $\mathbf{x}_A = (x_{l_1}, ..., x_{l_{|A|}})$ the vector of input variables with indices in $A$. When using Sobol indices based GSA, it is possible to measure the interactions between variables using ANOVA decomposition. For HSIC, the corresponding decomposition property (which is not ANOVA since HSIC does not measure the variance) can be stated as follows:

**Property 1** (Orthogonal decomposition property). *The orthogonal decomposition property is fulfilled if:*

$$HSIC(\mathbf{x}, \mathbf{y}) = \sum_{A \subseteq \{1,...,n\}} HSIC_A, \tag{3}$$

*where each term $HSIC_A$ is given by*

$$HSIC_A = \sum_{B \subseteq A} (-1)^{|A|-|B|} HSIC(\mathbf{x}_B, \mathbf{y}),$$

*and $HSIC(\mathbf{x}_B, \mathbf{y})$ is defined as in equation (1) with kernels $l$ and $k_A$.*

In Appendix A, we introduce an example to illustrate why this property is necessary in order to properly assess the importance of interactions. This property was lacking for dependence measure-based sensitivity analysis until the work of [9], which shows that when using HSIC, a specific choice of kernel $k$ can enable this decomposition. For any choice of $l$ and characteristic univariate kernel $k$, it is possible to obtain the orthogonal decomposition property by defining the input kernel $k_A$ such that

$$k_A(\boldsymbol{x}_A, \boldsymbol{x}'_A) = \prod_{i \in A}(1 + k_0(x_i, x'_i)), \quad s.t. \quad k_0(x, x') = k(x, x') - \frac{\int k(x,t)dP(t) \int k(x',t)dP(t)}{\int \int k(s,t)dP(s)dP(t)} \tag{4}$$

These conditions can be stringent, especially the right one, which implies to compute integrals (analytically or empirically). It can be non trivial for continuous input distributions $p_x$ and classical kernel choices such as Radial Basis Function (RBF) of bandwidth $\sigma$, $k(x, x') = \exp(\|x - x'\|/2\sigma^2)$. This condition is alleviated when using discrete input variables of known densities, for which the integrals can be computed analytically. In particular, in this work, we rely on Proposition 1:

**Proposition 1.** *Let $x_i$ be a Bernoulli variable of parameter $p = \frac{1}{2}$, and $\delta(x = x')$ the Dirac kernel. Then the following kernel $k_A$ satisfies the decomposition property:*

$$k_A(\boldsymbol{x}_A, \boldsymbol{x}'_A) = \prod_{i \in A}(1 + k_0(x_i, x'_i)), \quad s.t. \quad k_0(x, x') = \delta(x = x') - \frac{1}{2}. \tag{5}$$

The proof is in Appendix A. As a practical consequence, if we sample binary masks from a Bernoulli variable of parameter $p = 1/2$, i.e. $M_i \sim B(p)$ for $i \in \{1, ..., d\}$, and use the kernel defined in equation (5), we can assess not only the importance of each patch in the image but also the importance of the interactions between patches. It allows to go beyond classical attribution methods and reveal areas of the image that are only important in interaction with other areas, i.e. that affect the output only when both areas are perturbated at the same time. Concretely, for two image patches indexed by $i$ and $j$, the interaction HSIC, $\mathcal{H}_{i \times j}$, can be obtained with [9]:

$$\mathcal{H}_{i \times j}^p = \mathcal{H}_{(M_i, M_j), \mathbf{y}}^p - \mathcal{H}_{M_i, \mathbf{y}}^p - \mathcal{H}_{M_j, \mathbf{y}}^p. \tag{6}$$

We insist on the fact that if the decomposition property is not valid, subtracting $\mathcal{H}^p_{M_i,\mathbf{y}}$ and $\mathcal{H}^p_{M_j,\mathbf{y}}$ to $\mathcal{H}^p_{(M_i,M_j),\mathbf{y}}$ does not ensure that we assess the importance of the interactions only. Traces of the independent importance of $M_i$ and $M_j$ may remain in the obtained metric. Some qualitative benefits of such a property are illustrated in Section 4.4. This decomposition holds for any choice of kernel $l : \mathcal{Y} \to \mathcal{G}$. Therefore, in the following, we use the RBF kernel, with the common practice of choosing the bandwidth as the median of the output [8, 20, 48].

## 3.4 Sample efficiency of HSIC estimator

Several types of metrics are classically used for sensitivity analysis. The most famous one, Sobol indices [47] and its variants [16, 5] classically require $p^2$ model evaluations [23] to reach an estimation error of $\mathcal{O}(\frac{1}{\sqrt{p}})$. Recent design of experiments managed to reduce this requirement to $p \times (d+2)$ [39] (Theorem 1), but with the increase in complexity of state-of-the-art architectures and the high dimensionality of inputs (although mitigated by the use of $d$ patches instead of all pixels), it can still be cumbersome. Despite this drawback, [11] uses Sobol indices to obtain explanations and still achieves execution time improvement compared to other state-of-the-art attribution methods, which are even less efficient in terms of samples requirements.

HSIC is much less expensive to estimate than Sobol indices: for a same estimation error of $\mathcal{O}(\frac{1}{\sqrt{p}})$, $p$ forward passes are needed instead of $p \times (d+2)$ [20] (Theorem 1). This allows using far fewer samples to obtain relevant explanations, thereby dramatically increasing the efficiency of the method compared to previous black-box approaches. This huge advantage is empirically illustrated in Section 4.2, where we demonstrate that our method defines a new standard in terms of efficiency for black-box attribution methods. It even bridges the efficiency gap between black-box and white-box approaches.

## 3.5 Implementation of the method

A summary of the whole attribution method is provided in Algorithm 1. The computation of $\mathcal{H}^p_i$ is $O(p^2)$ but it is possible to vectorize it using any library optimized for tensor operations (e.g. tensorflow). As a result, the computation time of $\mathcal{H}^p_i$ is negligible compared to that of the $p$ forward passes. Furthermore, we implemented a sampling based on Latin Hypercube Sampling [30], a Quasi-Monte Carlo (QMC) method designed to efficiently fill the input space in Monte Carlo integration. Once the grid of $\mathcal{H}^p_i$ is obtained, we use a bilinear upsampling to be able to apply it to the image.

---

**Algorithm 1** Explanations using HSIC-based sensitivity analysis as attribution method

---

1: **Inputs:** $d$ the dimension of the masks, $p$ the number of forward pass, $\mathbf{X}$ an input image.
2: Sample $p$ binary masks $\{\mathbf{M}^{(1)}, ..., \mathbf{M}^{(p)}\}$ using LHS.
3: Compute the perturbed inputs $\{\boldsymbol{\tau}(\mathbf{X}, \mathbf{M}^{(1)}), ..., \boldsymbol{\tau}(\mathbf{X}, \mathbf{M}^{(p)})\}$
4: Compute the predictions $\{\mathbf{y}^{(1)}, ..., \mathbf{y}^{(p)}\} = \{\boldsymbol{f}(\boldsymbol{\tau}(\mathbf{X}, \mathbf{M}^{(1)})), ..., \boldsymbol{f}(\boldsymbol{\tau}(\mathbf{X}, \mathbf{M}^{(p)}))\}$
5: **for** $i \in \{1, ..., d\}$ **do**
6:     Compute $\mathcal{H}^p_i$ using equation (2) and assign this value to the $i$-th patch of the input image.

---

# 4 Experiments

This section showcases the benefits of our approach compared to other attribution methods. These benefits are threefold. First, the computational cost of HSIC attribution method is significantly lower than previous state-of-the-art methods, even bridging the performance gap between black-box and white-box methods. Second, our method improves state-of-the-art for several fidelity metrics, for black-box as well as white-box methods. Finally, the orthogonal decomposition property of HSIC allows to go beyond usual attribution methods and assess interactions between image patches.

In Section 4.1, we compute explanations of the predictions in the ILSVRC-2012 [10] classification task (ImageNet), for four common architectures, namely MobileNet [40], ResNet50 [22], EfficientNet [53] and VGG16 [44]. Then, we compare those explanations with these of other state-of-the-art black-box and white-box attribution methods in terms of fidelity and efficiency. In Section 4.2, we investigate the convergence of our method by measuring its correlation with a high sample estimator

| | Method | *ResNet50* | *VGG16* | *EfficientNet* | *MobileNetV2* | Exec. time (s) |
|---|---|---|---|---|---|---|
| **Del. (↓)** | | | | | | |
| White-box | Saliency [43] | 0.158 | 0.120 | 0.091 | 0.113 | 0.360 |
| | Grad.-Input [42] | 0.153 | 0.116 | 0.084 | 0.110 | 0.023 |
| | Integ.-Grad. [52] | 0.138 | **0.114** | **0.078** | 0.096 | 1.024 |
| | SmoothGrad [45] | 0.127 | 0.128 | 0.094 | **0.088** | 0.063 |
| | GradCAM++ [41] | **0.124** | 0.125 | 0.112 | 0.106 | 0.127 |
| | VarGrad [41] | 0.134 | 0.229 | 0.224 | 0.097 | 0.097 |
| Black-box | LIME [37] | 0.186 | 0.258 | 0.186 | 0.148 | 6.480 |
| | Kernel Shap [29] | 0.185 | 0.165 | 0.164 | 0.149 | 4.097 |
| | RISE [32] | 0.114 | 0.106 | 0.113 | 0.115 | 8.427 |
| | Sobol [11] | 0.121 | 0.109 | 0.104 | 0.107 | 5.254 |
| | $\mathcal{H}_i^p$ eff. (ours) | **0.106** | **0.100** | **0.095** | **0.094** | **0.956** |
| | $\mathcal{H}_i^p$ acc. (ours) | **0.105** | **0.099** | **0.094** | **0.093** | 1.668 |
| **Ins. (↑)** | | | | | | |
| White-box | Saliency [43] | 0.357 | 0.286 | 0.224 | 0.246 | 0.360 |
| | Grad.-Input [42] | 0.363 | 0.272 | 0.220 | 0.231 | 0.023 |
| | Integ.-Grad. [52] | 0.386 | 0.276 | 0.248 | 0.258 | 1.024 |
| | SmoothGrad [45] | 0.379 | 0.229 | 0.172 | 0.246 | 0.063 |
| | GradCAM++ [41] | 0.497 | **0.413** | **0.316** | 0.387 | 0.127 |
| | VarGrad [41] | **0.527** | 0.241 | 0.222 | **0.399** | 0.097 |
| Black-box | LIME [37] | 0.472 | 0.273 | 0.223 | 0.384 | 6.480 |
| | Kernel Shap [29] | 0.480 | 0.393 | 0.367 | 0.383 | 4.097 |
| | RISE [32] | **0.554** | **0.485** | **0.439** | **0.443** | 8.427 |
| | Sobol [11] | 0.370 | 0.313 | 0.309 | 0.331 | 5.254 |
| | $\mathcal{H}_i^p$ eff. (ours) | 0.470 | 0.387 | 0.357 | 0.381 | **0.956** |
| | $\mathcal{H}_i^p$ acc. (ours) | 0.481 | 0.395 | 0.366 | 0.392 | 1.668 |

Table 1: **Deletion** and **Insertion** scores obtained on 1,000 ImageNet validation set images (For Deletion, lower is better and for Insertion, higher is better). The execution times are averaged over 100 explanations of ResNet50 predictions with an RTX Quadro 8000 GPU. The first and second best results are **bolded** and underlined.

and comparing it with RISE [32] and Sobol [11]. In the remaining sections, we conduct additional experiments that show the versatility of our method. In Section 4.3, we evaluate HSIC attribution method to explain object detection on COCO dataset [28] with YOlOv4 [35]. We conclude the experiments with Section 4.4, where we showcase the use of the HSIC orthogonal decomposition property to assess interactions between image patches.

## 4.1 Fidelity of classification explanations

In this section, we evaluate the fidelity of the explanations with three fidelity metrics. The first, Deletion [32], assumes that the more faithful an explanation is, the quicker the prediction score should drop when pixels that are considered important are shut down. The second one, Insertion [32], instead adds pixels on a baseline image, starting with pixels that are associated with the highest importance scores of the explanation. Finally, $\mu$Fidelity [4] creates random pixels subsets which are assigned a baseline value and measure the correlation between the drop in the score and the importance of the explanation. Those metrics are further described in Appendix F.

In Table 1, we report the results of several different attribution methods for explaining the classification of MobileNet [40], ResNet50 [22], EfficientNet [53] and VGG16 [44] on 1000 images sampled from the ImageNet validation dataset. The models used for the experiments have been accessed from tensorflow [1] with the keras API [6]. We introduce two variants of our method, $\mathcal{H}_i^p$ eff. and $\mathcal{H}_i^p$ acc. The words "eff" and "acc" stand for efficient and accurate because we use $p = 764$ and $p = 1536$ samples, respectively. We use our method with a grid size of $7 \times 7$ ($d = 49$). To evaluate the different methods, we use the Xplique [12], a library dedicated to explainability. For black-box and white-box methods, we **bold** the best result and underline the second. When the differences between some methods are not statistically significant, we highlight both. Note that for $\mu$Fidelity, the estimation

variance is high (typically about $20\%$), so we only use the bold notation and leave the Table in Appendix B. The exact error bars are also left in the Appendix to make the presentation lighter.

For the Deletion metric, our method obtains the best results among the tested black-box methods for all the architectures in both its efficiency and accurate variants. Except for EfficientNet, we even beat all tested white-box methods. RISE is still the best of black-box methods for the Insertion metric, but the accurate variant is systematically second. Besides, our methods are among the best of both black-box and white-box methods.

While HSIC is systematically better in Deletion, we can note that RISE overshadows it in Insertion. This could be explained by how these metrics are constructed. Deletion and Insertion metrics consist in measuring Area Under the Curve (AUC) of scores that respectively decrease and increase when deleting and adding patches, starting from a baseline image (see Appendix F for a detailed definition). Since Deletion measures a drop in the score starting from the original image, the faster the score drops, the better the metric. Hence, Deletion will favor methods that sharply identify important regions. On the contrary, since Insertion starts from an arbitrary baseline image, if the explanation map is more spread out, more relevant secondary information will be added, so the score will be better. As we can see in the maps of Appendix C, RISE saliency maps are way more spread out than HSIC's, which are sharper. It may explain why RISE is better in the Insertion benchmark and why HSIC attribution method dominates the Deletion benchmark. We provide additional quantitative examples to illustrate this link in Appendix F. Note that even if RISE dominates Insertion, it is far behind in Deletion. This is not the case for HSIC, which is still competitive in Insertion while dominating Deletion.

These results, as such, are already satisfactory. But it goes even further: we obtained these results with far fewer forward passes than other state-of-the-art black-box attribution methods. With the efficient variants, competitive results are obtained more than $8$ times faster than RISE, the current standard of black-box attribution methods. It improves on Sobol, a recent and promising attribution method that was already branded as more efficient, by a factor $5$. The time improvement factors for the accurate variant of our method are still very appealing (5 and 3). We even beat the execution time of Integrated Gradients white-box method [52], a popular and successful white-box method. The efficiency of HSIC attribution method is investigated more thoroughly in the next section.

## 4.2 Estimator efficiency

The advantage of black-box attribution methods lies in providing explanations without access to the model's internal state or the gradients. However, this advantage comes at a cost since many forward passes are needed to obtain meaningful explanations. This cost is all the more constraining since recent architectures are increasingly heavy in terms of computational time.

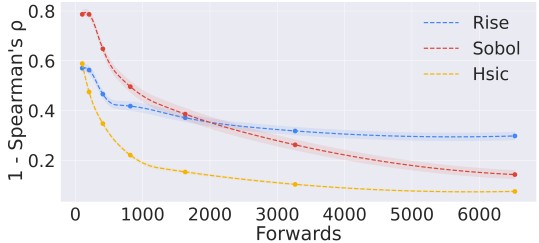

Figure 2: Spearman rank correlation of HSIC, Sobol and RISE attribution methods explanations on 100 ImageNet validation images with an "asymptotical" explanation based on $13,000$ samples.

Therefore, it is critical for such attribution methods to use as few forward passes as possible. Results reported in Section 4.1 attest that our approach based on $\mathcal{H}_i^p$ shines in that regard, and we refer to this section for more comments. It motivates us to study the efficiency of our method further. To that end, we compute an "asymptotical"[1] explanation with $13,000$ forward passes, for HSIC, Sobol and RISE attribution methods, on 100 images of ImageNet validation set.

We apply the three methods on EfficientNet with $d = 7 \times 7$ masks and image patches, like in [32] and [11]. We then compute explanations for an increasing number of forward passes and compare the obtained explanation with the baseline "asymptotical" explanation. We use Spearman rank correlation [49] like theoretically and empirically argued in [19, 2, 54, 13]. This experiment allows

---

[1]This explanation is not theoretically asymptotical (hence the quotation marks), but we use this designation because it is obtained with a very high number of forward passes

comparing the convergence speed of our method with RISE and Sobol. The curves are plotted in Figure 2 and show that our method, HSIC converges much faster than RISE and Sobol.

## 4.3 Explanation of object detection

Explaining model's predictions is more challenging for object detection than for image classification. Indeed, recent object detection models usually predict three pieces of information: localization (bounding box corners), objectness score (probability that a bounding box contains an object of any class), classification information (probability of each different possible class). Recently, it has been demonstrated that it was possible to use attribution methods to explain object detection by constructing a score aggregating for the previous information. In [33] the authors combine *intersection over union* for localization, *cosine similarity* for the class probability, and focus on high objectness areas. As a result, they can use RISE to explain the object detection using this score as the output of the model.

In this section, we test our method for explaining the object detections of YOLOv4 [35] on COCO dataset [28] compared to the approach presented in [33], D-RISE. We also compare the explanations with these of Kernel Shap [29], another black-box attribution method. The explanations for $1,000$ validation images are evaluated with the Deletion, Insertion, and $\mu$Fidelity metrics. This experiment is time-consuming, so we use $\mathcal{H}_i^p$ eff. and $5000$ samples for D-RISE and Kernel Shap.

| Method | Deletion ($\downarrow$) | Insertion ($\uparrow$) | $\mu$Fidelity ($\uparrow$) | Exec. time (s) |
|---|---|---|---|---|
| D-RISE [33] | 0.074 | 0.634 | 0.442 | 155 |
| Kernel Shap. [29] | **0.070** | 0.646 | 0.476 | 192 |
| $\mathcal{H}_i^p$ (ours) | 0.088 | **0.658** | **0.568** | **34** |

Table 2: Fidelity metrics obtained from explanations of YOLOv4 object detections on $1,000$ images of COCO validation data set. Execution times are averaged on the $1,000$ images on RTX 3080 GPUs.

Even if our method is not the best for Deletion, it is for Insertion and $\mu$Fidelity, and more importantly, it is $5$ and $6$ times faster than D-RISE and Kernel Shap. Figure 3 displays visualizations of object detection explanations. While the first images show a standard detection explanation, the rightmost one is more interesting since it emphasizes an error of the object detector. The model identifies a zebra as a cat, and our method manages to explain this error by emphasizing the cat at the bottom right corner of the image. Note that we did not obtain such an explanation with D-RISE, even with a high sample number and different grid sizes - visualizations can be found in Appendix C).

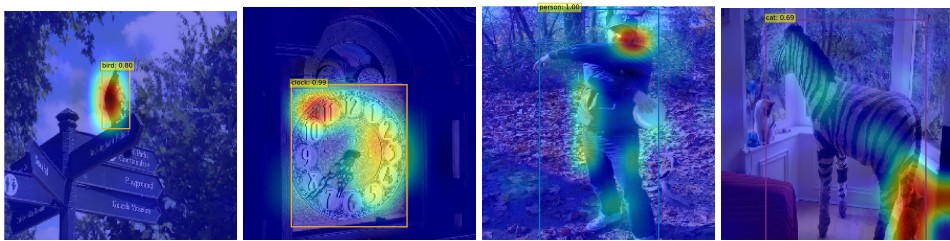

Figure 3: Visualisations of object detection explanations. The first three images show standard explanations, while the bottom right explains a misclassification. The zebra has been detected as a cat, and our method manages to explain why by emphasizing the cat at the bottom right corner.

## 4.4 Finding spacial interactions in the model

Usual attribution methods provide explanations in the form of heat maps that assign each pixel (or patch) an importance score. However, the scope of such explanations is limited since the reason for a prediction may not be explained only by the single importance of independent patches. In [11], the authors use Sobol total indices that account for the importance of a patch in interaction with all other patches, but they cannot localize the interactions. Thanks to its orthogonal decomposition property, our HSIC-based attribution method is able not only to assess the importance of each patch, but also the importance of interactions between specific patches by computing the HSIC of the joint patches

and subtracting the contribution of each patch taken independently[2]. It is then possible to identify regions of the image that affect the output only when both areas are perturbated jointly.

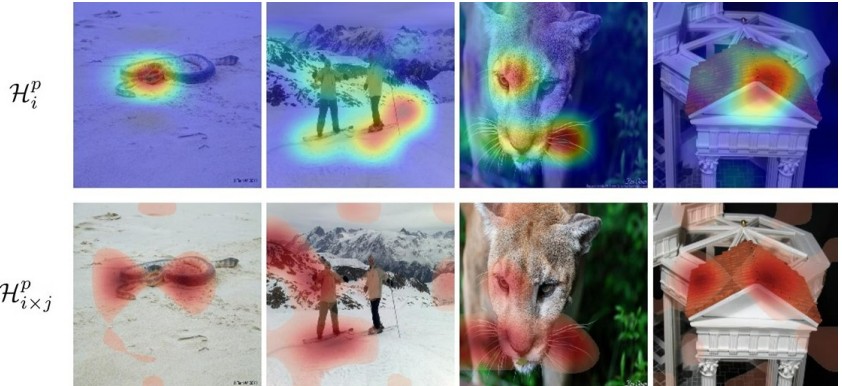

Figure 4: Upper row: $\mathcal{H}_i^p$, estimated HSIC for each image patch. Bottom row: highest $\mathcal{H}_{i \times j}^p$ between images patches. For the classification of the puma, the eye / forehead is the most important independent patch, but the right and left mustaches' interactions are even more important.

We illustrate this property in Figure 4. For each image, we computed all the possible interactions between patches and reported the most important ones. Note that not all images exhibit significant interactions, so we performed a qualitative selection for pedagogical purposes. On the upper row, we plot usual heatmaps that traduce the importance of each patch taken independently. On the bottom row, important interactions are represented in red. We can see that the middle part of the snake body is not the only important element in the picture, that one part of the mountain interacts with one of the skiers, so do the mustaches of the puma and two corners of the roof. Note that for each image, the maximum values of $\mathcal{H}_i^p$ are $40.6, 11.4, 8.2$ and $18.8$ and the maximum values of $\mathcal{H}_{i \times j}^p$ are $19.6, 6.6, 9.2$ and $6.3$ respectively. Thanks to the orthogonal decomposition property, these metrics can be compared and we can deduce that some interactions are as significant as some important independent patches. For the image with a puma, the interactions between the mustaches are even more important than the eye / forehead for identifying the animal ($\mathcal{H}_{i \times j}^p = 9.2$ when $i$ and $j$ are the right and left mustaches and $\mathcal{H}_{iff}^p = 8.2$ when $i$ is the eye / forehead).

## 5 Conclusion

We have introduced a new attribution method based on a dependence measure, Hilbert-Schmidt Independence Criterion, which leverages representation capabilities of Reproducing Kernel Hilbert Spaces, thus being able to capture complex information. This attribution method is black-box, so it is applicable even when the implementation of the neural network to explain is not available. Nonetheless, it alleviates the computational burden of traditional black-box methods, improving on the state-of-the-art of both black-box and white-box attribution methods while being closer to the latter than the former in terms of efficiency. In addition, we showed how the rich framework of RKHS could be used to assess and localize interactions between pairs of patches of the input image that are relevant for explaining the output. We hope that the introduced framework will open up research avenues for attribution methods beyond traditional pixel-wise or patch-wise explanations.

## Acknowledgements

This work has benefited from the AI Interdisciplinary Institute ANITI, which is funded by the French "Investing for the Future – PIA3" program under the Grant agreement ANR-19-P3IA-0004. The authors gratefully acknowledge the support of the DEEL project [3]. Part of this work was performed using HPC resources from CALMIP (Grant 2022-P22034). Additional support for computing hardware provided by Google via the TensorFlow Research Cloud (TFRC) program and by the Center for Computation and Visualization (CCV) at Brown University (NIH grant S10OD025181)

---

[2]With the decomposition property, it is also possible to obtain HSIC "total" indices, like for Sobol, but it did not bring significant qualitative or quantitative advantages.

[3]https://www.deel.ai/

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
