# A    On the Orthogonal Decomposition Property [1]

In this part, we state the orthogonal decomposition Property, motivate its importance with a pedagogical example, and finally prove Proposition 1, which enables the decomposition property in the context of HSIC attribution method.

## A.1    Orthogonal Decomposition Property

Let $\mathbf{x} = \{x_1, ..., x_n\} \in \mathcal{X}^n$ be a set of $n$ univariate random input variables. For any subset $A = \{l_1, ..., l_{|A|}\} \subseteq \{1, ..., n\}$, we denote $\mathbf{x}_A = (x_{l_1}, ..., x_{l_{|A|}})$ the vector of input variables with indices in $A$. Let $\mathbf{y}$ the random output variable defined by $\mathbf{y} = \boldsymbol{f}(\mathbf{x})$, $\mathcal{F}$ the RKHS defined by the kernel $k_A : \mathcal{X}^{|A|} \to \mathbb{R}$ and $\mathcal{G}$ the RKHS defined by the kernel $l : \mathcal{Y} \to \mathbb{R}$.

In [11], the author shows that for any choice of kernel $l$, if we respect some constraints on the kernel $k_A$, we can construct indices $HSIC(\mathbf{x}_A, \mathbf{y})$ that satisfy the following decomposition property.

**Property 2** (Decomposition property). *For any kernel $l$, the kernel $k_A$ satisfies the decomposition property if:*

$$HSIC(\mathbf{x}, \mathbf{y}) = \sum_{A \subseteq \{1,...,n\}} HSIC_A, \tag{7}$$

*where each term $HSIC_A$ is given by*

$$HSIC_A = \sum_{B \subseteq A} (-1)^{|A|-|B|} HSIC(\mathbf{x}_B, \mathbf{y}),$$

*and $HSIC(\mathbf{x}_B, \mathbf{y})$ is defined as in equation (1) with kernels $l$ and $k_A$.*

The constraints on the kernel $k_A$ are detailed in the main document and in the last section of this appendix. Before describing these constraints and how to fulfill them with Proposition 1, let us illustrate the importance of the property with a motivating, pedagogical example.

## A.2    Motivating example

In this section, we introduce a pedagogical example to motivate the interest in assessing the interactions and the importance of the Orthogonal Decomposition Property in that regard. Let $f : [0, 2]^3 \to \{0, 1\}$ such that

$$y = f(x_1, x_2, x_3) = \begin{cases} 1 & \text{if } x_1 \in [0, 1], x_2 \in [1, 2], x_3 \in [0, 1], \\ 1 & \text{if } x_1 \in [0, 1], x_2 \in [0, 1], x_3 \in [1, 2], \\ 0 & \text{otherwise.} \end{cases}$$

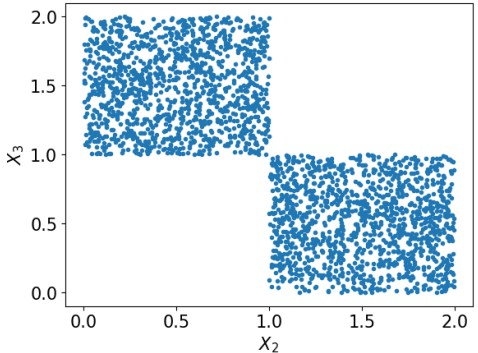

Figure 1: Input points $X_i = (x_{1,i}, x_{2,i}, x_{3,i})$ for which $f(X_i) = 1$ with respect to $x_2$ and $x_3$.

The relation between $x_2$, $x_3$ and the output of function $f$ is illustrated on Figure 7. Here $x_i$ is analogous to $M_i$. In that case, it is clear that $x_1$ is important to explain the output. However, assessing

the effect of $x_2$ and $x_3$ is more difficult. Given the definition of $f$; they are important to explain the output y, but it can be shown theoretically that $HSIC(x_2, y) = 0$ and $HSIC(x_3, y) = 0$ [34]. This motivates to assess the interactions between input variables. One way to retrieve the information that $x_2$ and $x_3$ are important is to assess $HSIC(x_{2,3}, y)$, where $x_{2,3} = (x_2, x_3)$

One could assess $HSIC(x_{2,3}, y)$ without any constraints on the kernel $k$, and the obtained value for $HSIC(x_{2,3}, y)$ would indeed be $> 0$ . However, by doing so, we would also obtain that $HSIC(x_{1,2}, y) > 0$ and $HSIC(x_{1,3}, y) > 0$, whereas $x_1$ does not interact with $x_2$ and $x_3$, only because of the individual effect of $x_1$. We empirically illustrate this by assessing these metrics using the estimator of Eq. 2 with $p = 10000$, and kernels $k, l$ chosen as the Radial Basis Function (RBF). The results are found in Table 1 below and show that:

- $HSIC(x_2, y) \approx HSIC(x_3, y) \approx 0$
- $HSIC(x_{1,2}, y) \approx HSIC(x_{1,3}, y) > HSIC(x_{2,3}, y)$

| $HSIC(x_1, y)$ | $HSIC(x_2, y)$ | $HSIC(x_3, y)$ | $HSIC(x_{1,3}, y)$ | $HSIC(x_{1,2}, y)$ | $HSIC(x_{2,3}, y)$ |
|---|---|---|---|---|---|
| $1.79 \times 10^{-2}$ | $2.28 \times 10^{-6}$ | $9.63 \times 10^{-6}$ | $1.36 \times 10^{-2}$ | $1.36 \times 10^{-2}$ | $2.92 \times 10^{-3}$ |

Table 1: HSIC metrics with $k$ taken as RBF

In order to correctly assess the pairwise interactions of input variables $x_1$ and $x_2$, one has to remove the individual effect of each variable from the $HSIC(x_{1,2}, y)$. The orthogonal decomposition property [11] allows to do so by simply computing $HSIC_{inter}(x_{1,2}, y)$ as

$$HSIC_{inter}(x_{1,2}, y) = HSIC(x_{1,2}, y) - HSIC(x_1, y) - HSIC(x_2, y)$$

**If the decomposition property does not hold, we are not guaranteed to fully remove the individual effect of** $x_1$ **and** $x_2$ **using the previous formula**. We estimate $HSIC_{inter}(x_{1,2}, y)$ when the kernel $k$ satisfies the decomposition property (in that case we choose a Sobolev kernel as in [1]), and when it does not, and show that the correct information of $HSIC_{inter}(x_{1,2}, y)$ is only retrieved when the decomposition property is satisfied. As previously, this is illustrated in the experiment, whose results are found in Table 2.

| | $HSIC(x_2, y)$ | $HSIC(x_3, y)$ | $HSIC(x_{1,3}, y)$ |
|---|---|---|---|
| $k$ Sobolev | $7.68 \times 10^{-6}$ | $2.83 \times 10^{-6}$ | $7.85 \times 10^{-4}$ |
| $k$ RBF | $-4.35 \times 10^{-3}$ | $-4.30 \times 10^{-3}$ | $2.91 \times 10^{-3}$ |

Table 2: HSIC metrics for assessing interactions, when $k$ satisfies (Sobolev) / does not satisfy (RBF) the orthogonal decomposition property

In that case, with $k$ satisfying the orthogonal decomposition property (Sobolev), we retrieve that $HSIC_{inter}(x_{1,2}, y) \approx HSIC_{inter}(x_{1,3}, y) \approx 0$ and $HSIC_{inter}(x_{2,3}, y)$ is significant. When $k$ does not satisfy the property (RBF), the values are not relevant (a negative value has no meaning since the metric is a distance)

## A.3   Proof of Proposition 1

To benefit from Property 2, the kernel $k_A$ must satisfy the following assumption [11]:

**Assumption 1.** *The kernel $k_A$ satisfies Property 2 if*

$$k_A(\boldsymbol{x}_A, \boldsymbol{x}'_A) = \prod_{i \in A} (1 + k_0(x_i, x'_i)),$$

*where*

$$k_0(x, x') = k(x, x') - \frac{\int k(x, t)dP(t) \int k(x', t)dP(t)}{\int \int k(s, t)dP(s)dP(t)}.$$

We now recall and prove the introduced Proposition 1 defined in Section 3.3.

**Proposition 1.** *Let* x *a Bernoulli variable of parameter* $p = 1/2$, *and* $\delta(x = x')$ *the dirac kernel such that* $\delta(x = x') = 1$ *if* $x = x'$ *and* $0$ *otherwise. Let* $k_0$ *be defined as in equation (4). Then, the kernel* $k_A$ *satisfies the decomposition property (Property 1) if it is defined according to Assumption 1 with*

$$k_0(x, x') = \delta(x = x') - \frac{1}{2}. \tag{8}$$

*Proof.* Let s and t be two iid random Bernoulli variables of parameter $p$ with probability density functions $p_s$ and $p_t$. We have that

$$\begin{cases} dP(s) = p_s(s)ds = \big(p\delta(s = 1) + (1 - p)\delta(s = 0)\big)ds \\ dP(t) = p_t(t)dt = \big(p\delta(t = 1) + (1 - p)\delta(t = 0)\big)dt. \end{cases}$$

Now, let's consider two Bernoulli variables x and x', two samples $x \sim$ x and $x' \sim$ x', and a kernel $k$ such that $k(x, x') = \delta(x = x')$.

- if x $\neq$ x'

$$\begin{cases} \int k(x, t)dP(t) \int k(x, s)dP(s) = p(1 - p) \\ \int \int k(s, t)dP(s)dP(t) = p^2 + (1 - p)^2 \end{cases}$$

- if x $=$ x' $= 0$

$$\begin{cases} \int k(x, t)dP(t) \int k(x, s)dP(s) = p^2 \\ \int \int k(s, t)dP(s)dP(t) = p^2 + (1 - p)^2 \end{cases}$$

- if x $=$ x' $= 1$

$$\begin{cases} \int k(x, t)dP(t) \int k(x, s)dP(s) = (p - 1)^2 \\ \int \int k(s, t)dP(s)dP(t) = p^2 + (1 - p)^2 \end{cases}$$

Therefore, since $p = \frac{1}{2}$,

$$\frac{\int k(x, t)dP(t) \int k(x', t)dP(t)}{\int \int k(s, t)dP(s)dP(t)} = \frac{1}{2},$$

so the kernel

$$k_0(x, x') = \delta(x = x') - \frac{1}{2}$$

satisfies the decomposition property 2.  $\square$

# B    Complete fidelity results

| | Method | *ResNet50* | *VGG16* | *EfficientNet* | *MobileNetV2* | Exec. time (s) |
|---|---|---|---|---|---|---|
| Del. ($\downarrow$) | | | | | | |
| White-box | Saliency [48] | $0.158 \pm 0.006$ | $0.120 \pm 0.005$ | $0.091 \pm 0.003$ | $0.113 \pm 0.004$ | 0.360 |
| | Grad.-Input [47] | $0.153 \pm 0.006$ | $\underline{0.116} \pm 0.004$ | $\underline{0.084} \pm 0.003$ | $0.110 \pm 0.004$ | 0.023 |
| | Integ.-Grad. [58] | $0.138 \pm 0.005$ | $\underline{0.114} \pm 0.004$ | $\mathbf{0.078} \pm 0.002$ | $\underline{0.096} \pm 0.004$ | 1.024 |
| | SmoothGrad [50] | $\underline{0.127} \pm 0.005$ | $0.128 \pm 0.005$ | $0.094 \pm 0.003$ | $\mathbf{0.088} \pm 0.003$ | 0.063 |
| | GradCAM++ [45] | $\mathbf{0.124} \pm 0.004$ | $\mathbf{0.105} \pm 0.003$ | $0.112 \pm 0.005$ | $0.106 \pm 0.005$ | 0.127 |
| | VarGrad [45] | $0.134 \pm 0.005$ | $0.229 \pm 0.007$ | $0.224 \pm 0.007$ | $\underline{0.097} \pm 0.004$ | 0.097 |
| Black-box | LIME [41] | $0.186 \pm 0.006$ | $0.258 \pm 0.008$ | $0.186 \pm 0.006$ | $0.148 \pm 0.006$ | 6.480 |
| | Kernel Shap [32] | $0.185 \pm 0.006$ | $0.165 \pm 0.006$ | $0.164 \pm 0.006$ | $0.149 \pm 0.006$ | 4.097 |
| | RISE [36] | $\underline{0.114} \pm 0.004$ | $\underline{0.106} \pm 0.004$ | $0.113 \pm 0.005$ | $0.115 \pm 0.004$ | 8.427 |
| | Sobol [13] | $0.121 \pm 0.003$ | $0.109 \pm 0.004$ | $\underline{0.104} \pm 0.003$ | $\underline{0.107} \pm 0.004$ | 5.254 |
| | $\mathcal{H}_i^p$ eff. (ours) | $\mathbf{0.106} \pm 0.003$ | $\mathbf{0.100} \pm 0.004$ | $\mathbf{0.095} \pm 0.003$ | $\mathbf{0.094} \pm 0.003$ | 0.956 |
| | $\mathcal{H}_i^b$ acc. (ours) | $\mathbf{0.105} \pm 0.003$ | $\mathbf{0.099} \pm 0.004$ | $\mathbf{0.094} \pm 0.003$ | $\mathbf{0.093} \pm 0.003$ | $\underline{1.668}$ |
| Ins. ($\uparrow$) | | | | | | |
| White-box | Saliency [48] | $0.357 \pm 0.009$ | $\underline{0.286} \pm 0.009$ | $0.224 \pm 0.008$ | $0.246 \pm 0.008$ | 0.360 |
| | Grad.-Input [47] | $0.363 \pm 0.010$ | $0.272 \pm 0.008$ | $0.220 \pm 0.009$ | $0.231 \pm 0.007$ | 0.023 |
| | Integ.-Grad. [58] | $0.386 \pm 0.010$ | $0.276 \pm 0.009$ | $\underline{0.248} \pm 0.008$ | $0.258 \pm 0.008$ | 1.024 |
| | SmoothGrad [50] | $0.379 \pm 0.010$ | $0.229 \pm 0.008$ | $0.172 \pm 0.006$ | $0.246 \pm 0.008$ | 0.063 |
| | GradCAM++ [45] | $\underline{0.497} \pm 0.010$ | $\mathbf{0.413} \pm 0.010$ | $\mathbf{0.316} \pm 0.009$ | $\underline{0.387} \pm 0.009$ | 0.127 |
| | VarGrad [45] | $\mathbf{0.527} \pm 0.010$ | $0.241 \pm 0.008$ | $0.222 \pm 0.007$ | $\mathbf{0.399} \pm 0.009$ | 0.097 |
| Black-box | LIME [41] | $0.472 \pm 0.010$ | $0.273 \pm 0.009$ | $0.223 \pm 0.007$ | $0.384 \pm 0.009$ | 6.480 |
| | Kernel Shap [32] | $\underline{0.480} \pm 0.010$ | $\underline{0.393} \pm 0.009$ | $\underline{0.367} \pm 0.008$ | $0.383 \pm 0.009$ | 4.097 |
| | RISE [36] | $\mathbf{0.554} \pm 0.010$ | $\mathbf{0.485} \pm 0.010$ | $\mathbf{0.439} \pm 0.009$ | $\mathbf{0.443} \pm 0.009$ | 8.427 |
| | Sobol [13] | $0.370 \pm 0.009$ | $0.313 \pm 0.009$ | $0.309 \pm 0.009$ | $0.331 \pm 0.009$ | 5.254 |
| | $\mathcal{H}_i^p$ eff. (ours) | $0.470 \pm 0.011$ | $0.387 \pm 0.010$ | $0.357 \pm 0.009$ | $0.381 \pm 0.009$ | 0.956 |
| | $\mathcal{H}_i^b$ acc. (ours) | $\underline{0.481} \pm 0.011$ | $\underline{0.395} \pm 0.011$ | $\underline{0.366} \pm 0.009$ | $\underline{0.392} \pm 0.009$ | $\underline{1.668}$ |

Table 3: **Deletion** and **Insertion** scores obtained on 1,000 ImageNet validation set images (For Deletion, lower is better and for Insertion, higher is better). The execution times are averaged over 100 explanations of ResNet50 predictions with a RTX Quadro 8000 GPU. The first and second best results are **bolded** and underlined.

| | Method | *ResNet50* | *VGG16* | *EfficientNet* | *MobileNetV2* | Exec. time (s) |
|---|---|---|---|---|---|---|
| White-box | Saliency [48] | $0.192 \pm 0.034$ | $0.092 \pm 0.035$ | $0.102 \pm 0.029$ | $\mathbf{0.172} \pm 0.030$ | 0.360 |
| | Grad.-Input [47] | $0.157 \pm 0.034$ | $0.066 \pm 0.029$ | $0.085 \pm 0.030$ | $0.116 \pm 0.029$ | 0.023 |
| | Integ.-Grad. [58] | $0.162 \pm 0.033$ | $0.073 \pm 0.029$ | $\mathbf{0.139} \pm 0.028$ | $\mathbf{0.157} \pm 0.030$ | 1.024 |
| | SmoothGrad [50] | $\mathbf{0.230} \pm 0.032$ | $0.087 \pm 0.030$ | $\mathbf{0.101} \pm 0.030$ | $0.126 \pm 0.028$ | 0.063 |
| | GradCAM++ [45] | $0.142 \pm 0.032$ | $\mathbf{0.143} \pm 0.032$ | $\mathbf{0.128} \pm 0.031$ | $0.131 \pm 0.029$ | 0.127 |
| | VarGrad [45] | $0.021 \pm 0.022$ | $0.022 \pm 0.020$ | $0.001 \pm 0.003$ | $0.101 \pm 0.032$ | 0.097 |
| Black-box | LIME [41] | $0.110 \pm 0.033$ | $0.015 \pm 0.032$ | $0.000 \pm 0.024$ | $0.055 \pm 0.031$ | 6.480 |
| | Kernel Shap [32] | $0.104 \pm 0.033$ | $0.068 \pm 0.034$ | $0.079 \pm 0.032$ | $0.051 \pm 0.031$ | 4.097 |
| | RISE [36] | $0.182 \pm 0.034$ | $0.099 \pm 0.034$ | $\mathbf{0.133} \pm 0.036$ | $\mathbf{0.123} \pm 0.031$ | 8.427 |
| | Sobol [13] | $\mathbf{0.230} \pm 0.034$ | $\mathbf{0.110} \pm 0.030$ | $\mathbf{0.141} \pm 0.034$ | $\mathbf{0.131} \pm 0.030$ | 5.254 |
| | $\mathcal{H}_i^p$ eff. (ours) | $\mathbf{0.202} \pm 0.034$ | $\mathbf{0.116} \pm 0.034$ | $\mathbf{0.154} \pm 0.035$ | $0.111 \pm 0.031$ | 0.956 |
| | $\mathcal{H}_i^b$ acc. (ours) | $0.187 \pm 0.035$ | $\mathbf{0.136} \pm 0.030$ | $\mathbf{0.155} \pm 0.035$ | $\mathbf{0.120} \pm 0.031$ | $\underline{1.668}$ |

Table 4: $\mu$**Fidelity** scores, obtained on 1,000 images from ImageNet validation set. Higher is better. The first and second best results are **bolded** and underlined. The execution times are averaged over 100 explanations of ResNet50 predictions with a RTX Quadro 8000 GPU.

# C    Additional visualizations on object detection explanations

## C.1    Visualizations

In this part we provide a sample of visualizations of object detection explanations for HSIC, RISE and KernelShap. HSIC seems more robust than the two other methods that are often blurry and sometimes fail. These images are taken from the 40 first images of COCO dataset. Out of transparancy, we

provide all the 40 first explanations in the github repository found at 

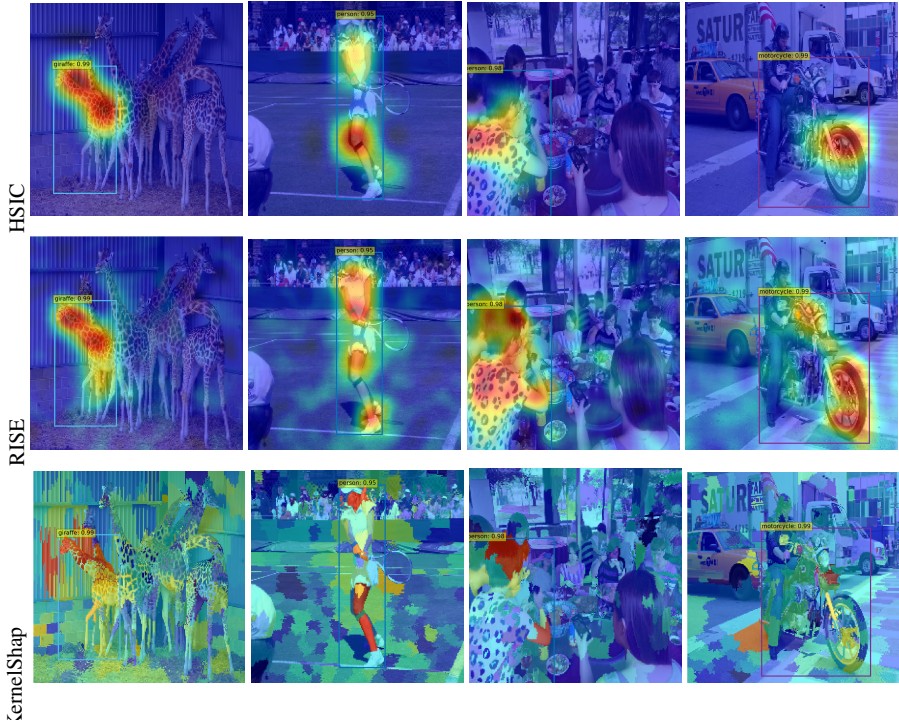

Figure 2: Visualisations of object detection explanations (1/2).

## C.2  Error explanations oh HSIC against RISE

In this section, we show explanations of RISE and KernelShap for the image where Yolov4 erroneously recognizes a cat instead of a zebra. HSIC manages to find an explanation for this error while both RISE and KernelShap fail, even for different grid sizes.

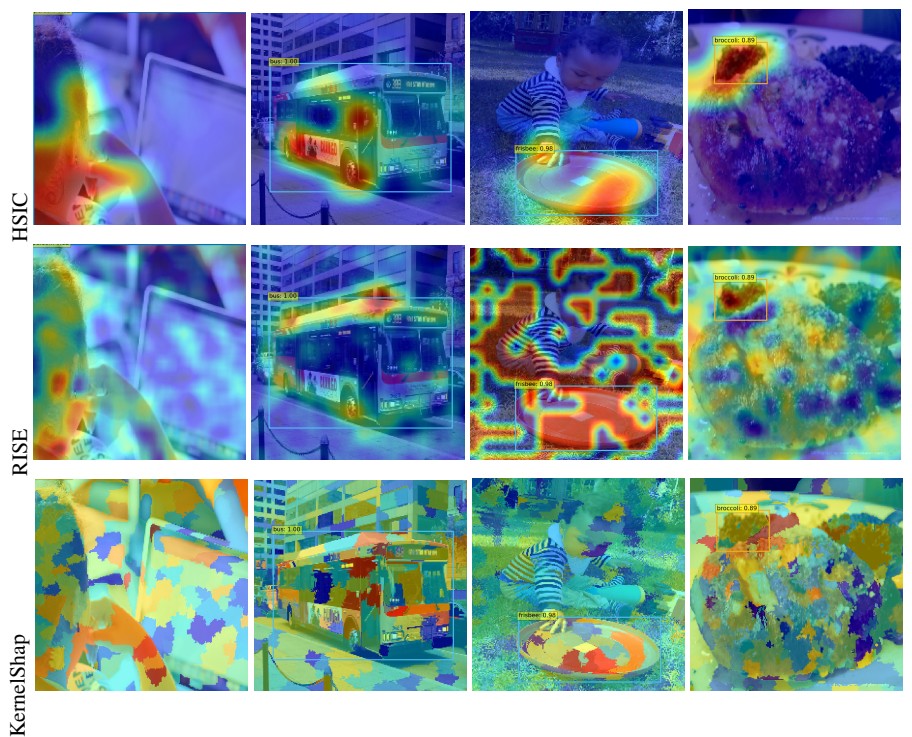

Figure 3: Visualisations of object detection explanations (2/2).

# D Additional visualizations of HSIC attribution method on ImageNet

HSIC

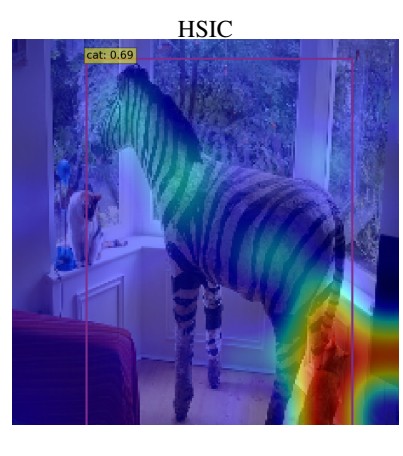

RISE

KernelShap

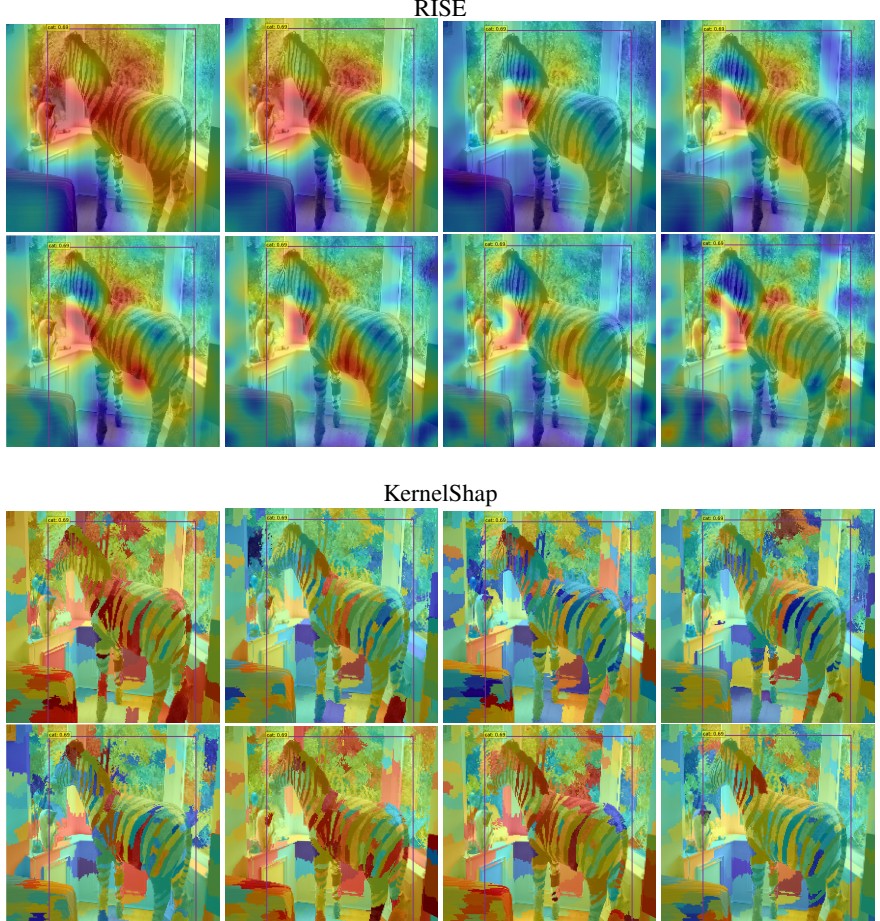

Figure 4: Visualizations of object detection explanations for a model error with HSIC method. Blurry explanations for different grid sizes with RISE and KernelShap.

# E   Attribution methods

In the following section, we give the formulation of the different attribution methods used in this work. The library used to generate the attribution maps is Xplique [14]. By simplification of notation,

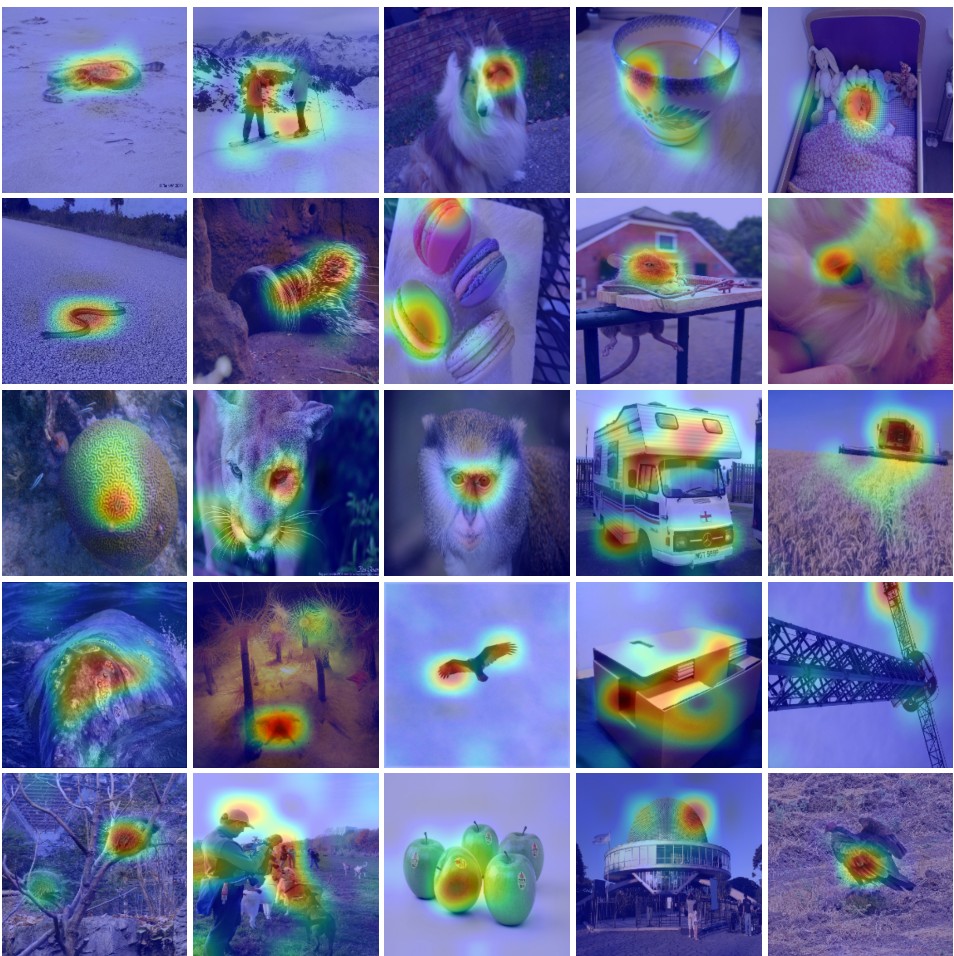

Figure 5: Explanations for ImageNet with HSIC eff.

we define $\boldsymbol{f}(x)$ the logit score (before softmax) for the class of interest (we omit $c$). We recall that an attribution method provides an importance score for each input variable $x_i$. We will denote the explanation functionnal mapping an input of interest $x = (x_1, ..., x_d)$ as $\boldsymbol{g}(x)$.

**Saliency** [48] is a visualization technique based on the gradient of a class score relative to the input, indicating in an infinitesimal neighborhood which pixels must be modified to most affect the score of the class of interest.

$$\boldsymbol{g}(x) = ||\nabla_x \boldsymbol{f}(x)||$$

**Gradient ⊙ Input** [46] is based on the gradient of a class score relative to the input, element-wise with the input, it was introduced to improve the sharpness of the attribution maps. A theoretical analysis conducted by [3] showed that Gradient ⊙ Input is equivalent to $\epsilon$-LRP and DeepLIFT [46] methods under certain conditions – using a baseline of zero and with all biases to zero.

$$\boldsymbol{g}(x) = x \odot ||\nabla_x \boldsymbol{f}(x)||$$

**Integrated Gradients** [58] consists of summing the gradient values along the path from a baseline state to the current value. The baseline $x_0$ used is zero. This integral can be approximated with a set

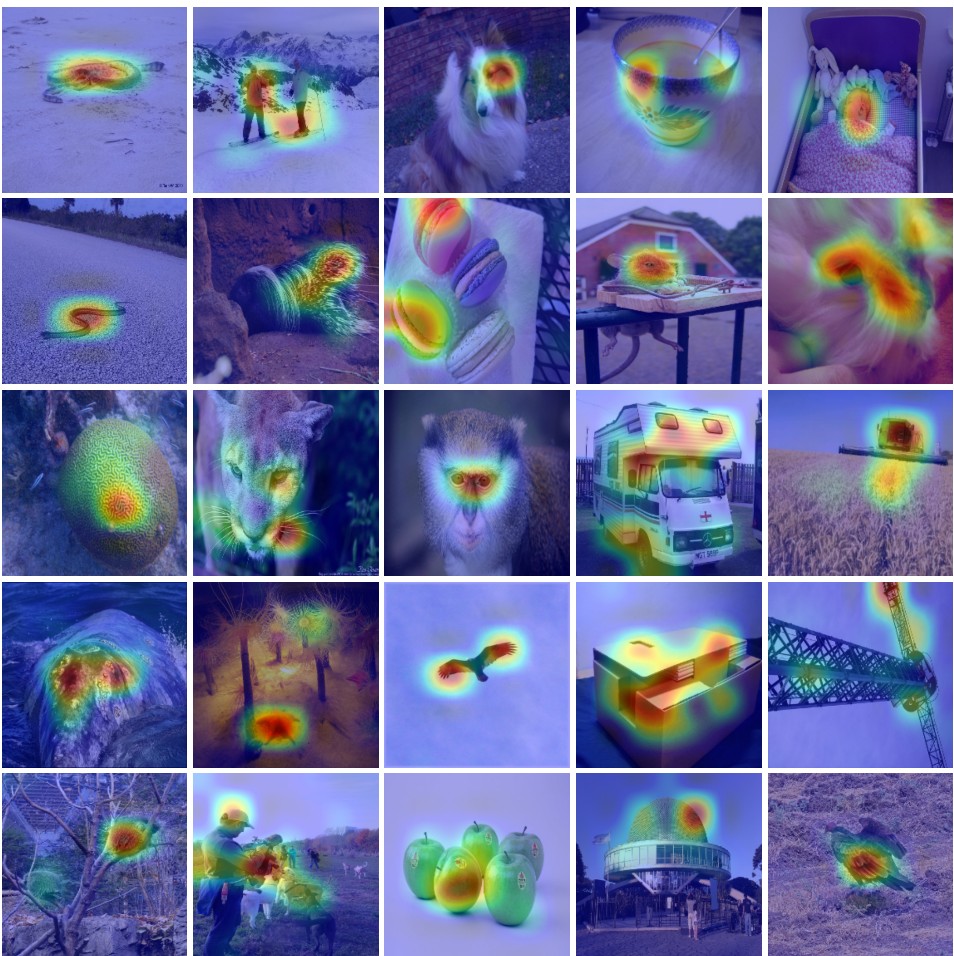

Figure 6: Explanations for ImageNet with HSIC acc.

of $m$ points at regular intervals between the baseline and the point of interest. In order to approximate from a finite number of steps, we use a Trapezoidal rule and not a left-Riemann summation, which allows for more accurate results and improved performance (see [53] for a comparison).

$$\boldsymbol{g}(x) = (x - x_0) \int_0^1 \nabla_x \boldsymbol{f}(x_0 + \alpha(x - x_0)))d\alpha$$

**SmoothGrad** [50] is also a gradient-based explanation method, which, as the name suggests, averages the gradient at several points corresponding to small perturbations (drawn i.i.d from an isotropic normal distribution of standard deviation $\sigma$) around the point of interest. The smoothing effect induced by the average help reducing the visual noise, and hence improve the explanations. The attribution is obtained by averaging after sampling $m$ points. For all the experiments, we took $m = 80$ and $\sigma = 0.2 \times (x_{\max} - x_{\min})$ where $(x_{\min}, x_{\max})$ being the input range of the dataset.

$$\boldsymbol{g}(x) = \mathbb{E}_{\delta \sim \mathcal{N}(0, \boldsymbol{I}\sigma)} (\nabla_x \boldsymbol{f}(x + \delta))$$

**VarGrad** [26] is similar to SmoothGrad as it employs the same methodology to construct the attribution maps: using a set of $m$ noisy inputs, it aggregate the gradients using the variance rather than the mean. For the experiment, $m$ and $\sigma$ are the same as Smoothgrad. Formally:

$$\boldsymbol{g}(x) = \underset{\delta \sim \mathcal{N}(0, \boldsymbol{I}\sigma)}{\mathbb{V}} (\nabla_x \boldsymbol{f}(x + \delta))$$

**Grad-CAM** [45] can only be used on Convolutional Neural Network (CNN). Thus we couldn't use it for the MNIST dataset. The method uses the gradient and the feature maps $\boldsymbol{A}^k$ of the last convolution layer. More precisely, to obtain the localization map for a class, we need to compute the weights $\alpha_c^k$ associated to each of the feature map activation $\boldsymbol{A}^k$, with $k$ the number of filters and $Z$ the number of features in each feature map, with $\alpha_k^c = \frac{1}{Z} \sum_i \sum_j \frac{\partial \boldsymbol{f}(x)}{\partial \boldsymbol{A}_{ij}^k}$ and

$$\boldsymbol{g} = \max(0, \sum_k \alpha_k^c \boldsymbol{A}^k)$$

As the size of the explanation depends on the size (width, height) of the last feature map, a bilinear interpolation is performed in order to find the same dimensions as the input. For all the experiments, we used the last convolutional layer of each model to compute the explanation.

**Grad-CAM++ (G+)** [7] is an extension of Grad-CAM combining the positive partial derivatives of feature maps of a convolutional layer with a weighted special class score. The weights $\alpha_c^{(k)}$ associated with each feature map are computed as follows:

$$\alpha_k^c = \sum_i \sum_j \left[ \frac{\frac{\partial^2 \boldsymbol{f}(x)}{(\partial \boldsymbol{A}_{ij}^{(k)})^2}}{2 \frac{\partial^2 \boldsymbol{f}(x)}{(\partial \boldsymbol{A}_{ij}^{(k)})^2} + \sum_i \sum_j \boldsymbol{A}_{ij}^{(k)} \frac{\partial^3 \boldsymbol{f}(x)}{(\partial \boldsymbol{A}_{ij}^{(k)})^3}} \right]$$

**Occlusion** [61] is a sensitivity method that sweeps a patch that occludes pixels over the images using a baseline state and uses the variations of the model prediction to deduce critical areas. For all the experiments, we took a patch size and a patch stride of $\frac{1}{7}$ of the image size. Moreover, the baseline state $x_0$ was zero.

$$\boldsymbol{g}(x)_i = \boldsymbol{f}(x) - \boldsymbol{f}(x_{[x_i=0]})$$

**RISE** [36] is a black-box method that consists of probing the model with $N$ randomly masked versions of the input image to deduce the importance of each pixel using the corresponding outputs. The masks $\boldsymbol{m} \sim \mathcal{M}$ are generated randomly in a subspace of the input space. For all the experiments, we use a subspace of size $7 \times 7$ and $\mathbb{E}(\mathcal{M}) = 0.5$.

$$\boldsymbol{g}(x) = \frac{1}{\mathbb{E}(\mathcal{M})N} \sum_{i=0}^{N} \boldsymbol{f}(x \odot \boldsymbol{m}_i) \boldsymbol{m}_i$$

## F   Evaluation

For the purpose of the experiments, three fidelity metrics have been chosen. For the whole set of metrics, $\boldsymbol{f}(x)$ score is the score after softmax of the models. We first describe these metrics and then discuss the trade-off between Deletion and Insertion scores.

### F.1   Definitions

**Deletion.**   [36] The first metric is Deletion, it consists in measuring the drop in the score when the important variables are set to a baseline state. Intuitively, a sharper drop indicates that the explanation method has well identified the important variables for the decision. The operation is repeated on the whole image until all the pixels are at a baseline state. Formally, at step $k$, with $\boldsymbol{u}$ the most important variables according to an attribution method, the Deletion$^{(k)}$ score is given by:

$$\text{Deletion}^{(k)} = \boldsymbol{f}(x_{[x_{\boldsymbol{u}}=x_0]})$$

We then measure the AUC of the Deletion scores. For all the experiments, the baseline state is fixed at $x_0 = 0$.

**Insertion.** [36] Insertion consists in performing the inverse of Deletion, starting with an image in a baseline state and then progressively adding the most important variables. Formally, at step $k$, with $\boldsymbol{u}$ the most important variables according to an attribution method, the Insertion$^{(k)}$ score is given by:

$$\text{Insertion}^{(k)} = \boldsymbol{f}(x_{[x_{\overline{\boldsymbol{u}}}=x_0]})$$

We then measure the AUC of the Deletion scores. The baseline is the same as for Deletion.

$\mu$**Fidelity** [5] consists in measuring the correlation between the fall of the score when variables are put at a baseline state and the importance of these variables. Formally:

$$\mu\text{Fidelity} = \underset{\substack{\boldsymbol{u} \subseteq \{1,\dots,d\} \\ |\boldsymbol{u}|=k}}{\text{Corr}} \left( \sum_{i \in \boldsymbol{u}} \boldsymbol{g}(x)_i, \boldsymbol{f}(x) - \boldsymbol{f}(x_{[x_{\boldsymbol{u}}=x_0]}) \right)$$

For all experiments, $k$ is equal to 20% of the total number of variables and the baseline is the same as the one used by Deletion.

### F.2  Trade-off between Insertion and Deletion

Deletion and Insertion metrics consist in measuring AUC of scores that respectively decrease and increase when deleting and adding patches, starting from a baseline image. Since the patches deleted/added are those that are the most important (in the sense of the tested attribution method), most of the score will come from the first patch deletions/additions. Using those different methods has two important consequences, detailed below.

**Deletion is preferable**    There is a key difference between those two evaluations that makes Deletion more suited to explanation evaluation than Insertion. In Deletion, since we start from the original image and sequentially delete patches, the score is tested in a region of the input image space that is close to the input image. On the contrary, Insertion starts from an arbitrary baseline (here, pure black image), which is far from the input image. It is likely that the value of the baseline has an undesired impact on the score for Insertion. That is why we tend to favor Deletion over Insertion.

**Some methods are more suited to Deletion or Insertion**    Since Deletion measures a drop in the score, the faster the score drops, the better the metric. Hence, Deletion will favor methods that sharply identify important regions. On the contrary, since Insertion starts from an arbitrary baseline image, if the explanation map is more spread out, more relevant secondary information will be added, so the score will be better. To illustrate this observation, in table 5 we show the value of Insertion and Deletion metrics for HSIC method and for different grid sizes, obtained after a grid search for MobileNetV2 on 1000 ImageNet validation images. The metrics are averaged over 27 runs (with a different number of samples and different samplers). Table 5 gives an idea of the trend of the evolution of Insertion and Deletion with respect to the grid size. As we can see, Deletion improves when the grid size increases, i.e. when the explanation map becomes sharper, and Insertion improves when the grid size decreases, i.e. when the map becomes more spread out.

| grid size | 5 | 6 | 7 | 8 | 9 | 10 |
|---|---|---|---|---|---|---|
| Insertion $\times 10^{-1}$ | 4.14 | 4.02 | 3.90 | 3.72 | 3.54 | 3.40 |
| Deletion $\times 10^{-1}$ | 1.01 | 0.97 | 0.94 | 0.93 | 0.92 | 0.90 |

Table 5: Result of a grid search for MobileNetV2

This trend also explains why RISE shines in the Insertion benchmark and why our HSIC attribution method dominates the Deletion benchmark. Indeed, as we can see in the maps of Appendix C, RISE saliency maps are way more spread out than HSIC's, which are sharper.

# G  Additional experiments on stability

In this section, we report the evolution of the Deletion score for HSIC, RISE, and Sobol with respect to the number of forward passes, with a Resnet50 on 100 Imagenet validation images.

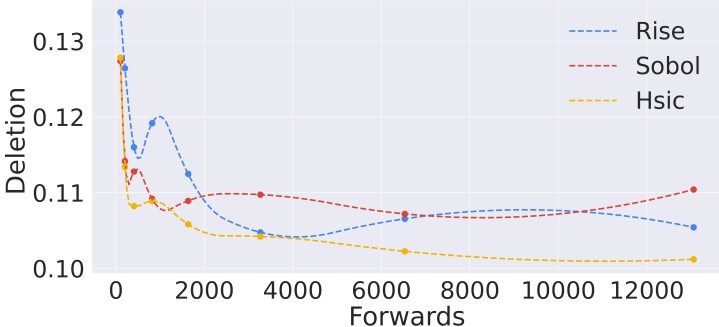

Figure 7: Deletion score for HSIC, RISE, and Sobol with respect to the number of forward passes

The scores for Sobol and RISE are less stable than for HSIC, which corroborates that HSIC attribution method can be used with fewer forward passes.