# OpenReview forum: "Making Sense of Dependence: Efficient Black-box Explanations Using Dependence Measure"
_NeurIPS.cc/2022/Conference — NeurIPS 2022 Accept_

### Official Review · Reviewer_eNpV · 2022-07-06

**Rating:** 6
**Confidence:** 3
**Soundness:** 3 good
**Presentation:** 2 fair
**Contribution:** 2 fair

**Summary:**

After rebuttal: The authors answered my questions. The ideas are interesting, but the improvement of the state-of-the-art is perhaps not that significant and the presentation could be improved (which the authors can probably do if accepted). I increased my rating to weak accept.

---


The authors propose a new attribution method based on the Hilbert-Schmidt Independence Criterion. The paper explains the technical background, defines and motivates the attribution method and proposes a simple sampling algorithm to approximate it. The authors also discuss how the new method can be better suited for capturing the joint effects of features. In the experiments, the paper evaluates 'fidelity', 'efficiency' and finding spatial interactions.

**Questions:**

The main points from my review above that could change my opinion are the following (please correct me if I am wrong):

1) The main motivation for the particular choice of the attribution method seems to be that it features a decomposition property that allows separting the joint effect of features from their individual effects. While this is theoretically interesting, it could be discussed and evaluated in more detail if it gives much better explanations than simply considering pairs of features that could then be evaluated by a standard method.

2) Previous results from the literature seem to suggest that the estimator applied here needs fewer samples. However, the discussion is somewhat disconnected from the actual algorithm and it would be good to be more explicit about which guarantees (please refer to the theorem number) from which paper can be applied here and why.

3a) Table 1 shows Deletion and Insertion results, but muFidelity seems to be missing. The proposed approach works best with respect to Deletion, but not with respect to Insertion, which is dominated by RISE. This indicates that there could be a tradeoff between Insetion and Deletion similar to Precision and Recall. Perhaps it's just a coincidence, but it would be good to discuss why the proposed method does better with respect to one measure, but not with respect to the other.

3b) I am also missing muFidelity in the table.

4) The runtime experiment is probably supposed to illustrate the convergence speed. While the graph intuitively indicates faster convergence, it should be discussed in more detail what it actually shows and why the experiment has been set up in this way. In particular, what is the "baseline asymptotical explanation"?


**Limitations:**

Yes

**Strengths And Weaknesses:**

The idea seems novel and is relevant for NeurIPS. The main motivation for the particular choice of the attribution method seems to be that it features a decomposition property that allows separting the joint effect of features from their individual effects. While this is theoretically interesting, it could be discussed and evaluated in more detail if it gives much better explanations than simply considering pairs of features that could then be evaluated by a standard method.  To get an 'efficient method' the authors use a sampling algorithm. Previous results from the literature seem to suggest that the estimator applied here needs fewer samples. However, the discussion is somewhat disconnected from the actual algorithm and it would be good to be more explicit about which guarantees (please refer to the theorem number) from which paper can be applied here and why.

The 'fidelity' evaluation is based on the measures Deletion, Insertion and muFidelity. They are explained in the appendix, which should be mentioned in the main part (I almost missed it). I suppose that Insertion is based on AUC symmetrical to Deletion, but it would be good to make this explicit). Table 1 shows Deletion and Insertion results, but muFidelity seems to be missing. The proposed approach works best with respect to Deletion, but not with respect to Insertion, which is dominated by RISE. This indicates that there could be a tradeoff between Insetion and Deletion similar to Precision and Recall. Perhaps it's just a coincidence, but it would be good to discuss why the proposed method does better with respect to one measure, but not with respect to the other. I am also missing muFidelity in the table. The runtime experiment is probably supposed to illustrate the convergence speed. While the graph intuitively indicates faster convergence, it should be discussed in more detail what it actually shows and why the experiment has been set up in this way. In particular, what is the "baseline asymptotical explanation"? Interactions are investigated by means of examples. The picked examples are intuitive. While such an evaluation by example seems always a bit weak, it does make sense that the joint effect of features can be more important than their individual effects.

Overall, the paper does not contain ground-breaking results, but shows improvements in some areas (Deletion, Runtime). There are some interesting ideas in the paper, but they should be discussed and evaluated in more detail in my opinion (e.g. feature interaction, runtime and approximation error guarantees). The code is available for reproducibility, although I did not have time to check it. At the moment, I consider this as a borderline paper.

Some comments

- I find the black-box/white-box distinction in this paper confusing. White-box sometimes refers to a model that is 'interpretable', while black-box refers to one that is not. Here, white-box just means that the model can be accessed, while black-box means that it cannot. Why not simply call it available/unavailable to avoid confusion? I am also not convinced that availability is really an issue in many cases. Of course, an end-user does not have access to the model, but in many applications, the explanation module should probably be part of the user interface. I may be wrong, and some concrete examples could convince me otherwise.

- Parts of the paper are very vague and convoluted. For example, lines 37-41 would be perhaps be easier to understand when just writing that perturbation methods rely on sampling and evaluating a sample can be expensive for recent neural network architectures that are growing larger.

- line 31: the paper refers to [18] for a statement about gradients, but the paper does not really seem to discuss the issue. Please clarify or delete.

---

> ### Author Response · Authors · 2022-08-01
> **Official Answer from Paper1305 Authors to Reviewer eNpV**
>
> Thank you very much for your detailed feedback. We address your comments and questions following the outline of your review.
> ## Comments
> * We agree that the terminology of black-box / white box attribution method may be confusing, but that this is how they are referred to in the recent literature (e.g. [3,4,5]). Hence, we prefer to keep this formulation for consistency with prior works.
> * Following your guidance, we have made some parts of the paper clearer.
> * The reference [18] is not used to discuss the gradients but to justify the fact that considering infinitesimal perturbations can be misleading.
> ## Questions
>
> **1.** We introduce an illustrating toy example in an openreview general comment above **"Motivating example for orthogonal decomposition property"**. This example clarifies the motivations for assessing the interactions and shows why the orthogonal decomposition property, granted by Proposition 1, is necessary to be able to do so.  It demonstrates that without this property, applying a standard method to a pair of patches would not actually measure the importance of interactions
>
> **2.** In the paper, we explicit that Sobol estimator theoretically needs $p \times (d+2)$ samples to be estimated with an error in $\mathcal{O}\Big( \frac{1}{\sqrt{p}}\Big)$, while HSIC only needs $p$ samples. These statements are referenced in [1] for Sobol, Theorem 1, and in [2] for HSIC, Theorem 1. To the best of our knowledge, they are the only methods that enjoy Theorems for their approximation guarantees. We add the mention of the theorems in the manuscript.
>
> **3.** As it is mentioned in the paper (l.260), $\mu$Fidelity Table has been postponed in the appendix because standard estimation errors were particularly high.
>
> ***"The proposed approach works best with respect to Deletion, but not with respect to Insertion, which is dominated by RISE. This indicates that there could be a tradeoff between Insertion and Deletion similar to Precision and Recall"***.
> Since Deletion measures a drop in the score, the faster the score drops, the better the metric. Hence, Deletion will favor methods that sharply identify important regions. On the contrary, since Insertion starts from an arbitrary baseline image, if the explanation map is more spread out, more relevant secondary information will be added, so the score will be better. As we can see in the maps of Appendix C, RISE saliency maps are way more spread out than HSIC's, which are sharper. It may explain why RISE is better in the Insertion benchmark and why HSIC attribution method dominates the Deletion benchmark (Section 4.1 Table 1). We provide additional quantitative examples to illustrate this link in Appendix F. Note that even if RISE dominates Insertion, it is far behind in Deletion. This is not the case for HSIC, which is still competitive in Insertion while dominating Deletion. We add those discussions in Appendix F.
>
> **4.** The experiment of Section 4.2, assessing the runtime of HSIC w.r.t. other methods, is inspired by the paper [3], which compares the convergence speed of Sobol with RISE. The convergence is assessed by computing the correlation of the explanations for an increasing number of samples (axis Forward in Figure 2) with what we call a "baseline asymptotical explanation". Ideally, this "baseline asymptotical explanation" should be the explanation obtained with $p \rightarrow +\infty$. Since we cannot obtain this baseline, we approximate it with a very high number of samples (way larger values used in practice), here 13,000. As we explain in the footnote of Section 4.2, we use quotations between "asymptotical" because it is not theoretically asymptotical
>
> [1] Making best use of model evaluations to compute sensitivity indices, Saltelli, Computer Physics Communications, 2002
>
> [2] Measuring Statistical Dependence with Hilbert-Schmidt Norms, Gretton et al, Proceedings of the 16th International Conference on Algorithmic Learning Theory, 2005
>
> [3] Look at the Variance!  Efficient Black-box Explanations with Sobol-based Sensitivity Analysis, T. Fel, R. Cadene, M. Chalvidal, M. Cord, D. Vigouroux, and T. Serre, NeurIPS 2021
>
> [4] Rise: Randomized input sampling for explanation of black-box models. V. Petsiuk, A. Das, and K. Saenko. In Proceedings of the British Machine Vision Conference (BMVC), 2018.
>
> [5] Black-box explanation of object detectors via saliency maps,  V. Petsiuk, R. Jain, V. Manjunatha, V. I. Morariu, A. Mehra, V. Ordonez, and K. Saenko. 2021 IEEE/CVF Conference on Computer Vision and Pattern Recognition (CVPR)

---

> > ### Comment · Reviewer_eNpV · 2022-08-08
> > **Acknowledgement**
> >
> > Thank you for the additional explanations. I will take them into account in the final discussion and will update my review accordingly.

---

### Official Review · Reviewer_QhZj · 2022-07-08

**Rating:** 4
**Confidence:** 3
**Soundness:** 3 good
**Presentation:** 2 fair
**Contribution:** 3 good

**Summary:**

The paper presents a new method for obtaining saliency maps that can explain a deep image classifier's predictions. The method is "black box" in the sense that it only relies on the ability to evaluate a model's predictions (obtain the predicted label "y" for any desired input "x"). It does not require any knowledge or access of the internals of the prediction function (e.g. the ability to compute gradients) as "white box" methods would.

The proposed method uses the Hilbert-Schmidt Independence Criterion (HSIC).
It assumes the selection of two kernel functions: "k" defines the similarity between two images (x and x') and "l" defines the similarity between two labels (y and y'). These kernels have associated feature embeddings.

Given p possible masks that perturb an original image x, and the associated outputs, the HSIC concretely computes (see Eq 2) the norm of the cross-covariance between the embeddings of perturbed images and embeddings of their predictions. The scalar quantity HSIC can be computed for each of the patches in an image.

They further describe how *interactions* between two patches in the same image can be given an HSIC score (Eq 5), which allows higher-order reasoning about importance (e.g. in an image of a face, perhaps each eye individually is not so important but the combination of the two eyes is).

Overall, the claimed contributions of the proposed approach are:
- 1) a new method based on HSIC
- 2) a derivation that shows an orthogonal decomposition property exists for a specific kernel
- 3) experiments showing the method is useful on ImageNet in terms of quality and speed
- 4) demonstration of versatility on object detection and in pairwise patch interactions


**Questions:**


### Q1: What was done to ensure fair comparison of timings in Table 1?

How many perturbations p are Sobol or RISE using? How was this selected?
Shouldn't an "efficient" and "accurate" version of both methods be also possible?


### Q2: In Table 2, why use 5000 samples for D-RISE but fewer for your method?

Again, seems like this choice will impact the speed of baselines.
How can we be sure this choice is fair?

### Q3: What does it mean for a method to score better on Insertion than Deletion?

RISE is notably better on that Insertion metric than others.
What's the intuition here? Are there ways to visually understand why that is?


Minor Presentation Issues
-------------------------
Please adjust for revision, but no need to discuss in rebuttal

- Lines 25-28 read awkwardly, consider revising for flow
- Fig 1 gives impression that x has a distribution, but the distribution is over binary mask, right?
- Lines 46-49: A distribution over what exactly? Over perturbed values? Over pixels observed in a region/patch?
- Lines 123-126: Why use the symbol x twice (italicized and not)? Please redefine
- Line 171: "networkss" typo

**Limitations:**

Limitations
-----------
There are likely a number of choices that could be very sensitive to overall performance that are under-explored here:
- number of patches used / size of the 'grid'
- choice of the kernel functions (both "k" and "l")
- choice of the lengthscale of the RBF kernel (maybe median is OK, but could you improve performance by optimizing?)

As a blackbox method, there may be cases where white box methods are possible and preferred, but this is adequately addressed by the paper.

The discussion of negative societal impact was a bit light, though unobjectionable (I agree that for a new technical method like this, it will share the same possible for misuse as related methods like Sobol or RISE).

**Strengths And Weaknesses:**


Strengths
---------
+ Speed ups of 5-8x over previous black-box attribution methods (Sobol and RISE) seem useful
+ The RKHS foundations of the proposed methods are technically interesting
+ Focus on pairwise interactions between patches is useful
+ Experiments compare to a variety of "state of the art" white and black box methods

Weaknesses
----------
- W1: Technical communication could be better, some key details are difficult to parse [see below]
- W2: Orthogonal decomposition property in particular is not well explained or justified
- W3: Qualitative visual examples are only shown for the presented method; difficult to assess relative value over baselines
- W4: Some of the visual explanations are puzzling: Why is a Puma's mouth more important than eyes? Why is only the number 11 important on the clock?


### W1: Technical communication issues

In Sec 3.1, there are several issues:

1) the definition of "n" versus "d" is not clearly established.
It is not clear how the elementwise product between x (a vector of size n) and M (a vector of size d) is well-defined.
It is not clear if "patches" or individual pixels are used.

2) the way that "baseline" value \mu is defined is not clear
Is this value the same for every pixel in image? How is it defined? is it learned?

In Sec 3.2, further issues:

3) It is difficult to do a dimensionality analysis of Eq 2
Looking at Eq 2, the trace of a product of pxp matrices should be a scalar.
But surely the HSIC is computed for each pixel/patch in the image, right?
This equation's presentation should be modified to more clearly indicate how you get a scalar for each pixel/patch of the saliency "heatmap"


4) Using symbol "x" here as a general random variable collides with readers' previous understanding of x as the input image.
I think you want to distinguish the two.... there isn't a distribution over the input image, instead there's a distribution over *perturbations* of this image.
For example, in Eq 4 you need x to be discrete, but this collides with reader's already formed assumption that x is a pixel.

### W2: Orthogonal decomposition is not clear

I understand at a high level that assessing the interaction between two patches in the image is important and it is not additive. But the technical details of the proposed "orthogonal decomposition" were not clear to me.

Lines 176-182 were particularly hard to read for me.
Why is the "sum of indices" equal to one?
Why does this imply something beneficial?

---

> ### Author Response · Authors · 2022-08-01
> **Official Answer from Paper1305 Authors to Reviewer QhZj (1/2)**
>
> Thank you for your detailed feedback. We have addressed the minor issues directly in the revised version. The answers are organized following the outline of your review.
> ## Weaknesses
> ### W1 - Technical communication issues
> #### Section 3.1
> 1. There is indeed a clash between dimensions $n$ and $d$. We solved this problem by adding another type of mask $M'$ constructed out of $M$, which is an upsampled version of $M$ to match the dimension $n$ of the input image.
> 2. For the baseline value $\mu$, we use a black image (with value 0).
> #### Section 3.2
> 1. Indeed, the obtained indices $\mathcal{H}_i$ are scalars that are computed for each individual image patch. We make it clearer in the description of Eq. 2.
> 2. We follow your suggestion and make the notations easier to read by denoting the input image by $X$ instead of $x$.
> ### W2 - Orthogonal decomposition is not clear
> We included a new pedagogical example in an openreview general comment above **"Motivating example for orthogonal decomposition property"**. This example shows the importance of assessing the interactions and how the orthogonal decomposition property is necessary to do so. Note also that a more detailed description of the orthogonal decomposition property can be found in Appendix A.
>
> The formulation ***"When using Sobol indices based GSA, the sum of the indices of all possible $\textnormal{\textbf{x}}_A$ is equal to 1"*** refers to the ANOVA / orthogonal decomposition property of Sobol indices, which are often normalized so that their sum is equal to 1. In the sensitivity analysis literature, this way of referring to the decomposition property is sometimes used to convey that the indices measure the independent (orthogonal) effects of variables. We plan to describe the decomposition property more thoroughly in the main document upon acceptance so that there is less confusion about these wordings.
> ### W3
> We already included visualizations of explanations for kernelShap and RISE in Appendix C.
> ### W4
> Attribution methods aim at explaining the output of a neural network, not finding human-level explanations. When the method emphasizes that a Puma's mouth is more important than its eyes, it only means that it is **from the perspective of the neural network**. In fact, there is a broad consensus that measuring the human-level plausibility of an explanation alone is not sufficient [1, 2]. The goal of explainability is to uncover the true underlying decision process of a model, not the consensus with a human explanation.
> ## Questions
>
> ***1. What was done to ensure fair comparison of timings in Table 1?***
> The numbers of perturbations used in RISE and Sobol are those of the original papers, i.e. 8000 for RISE and 4900 for Sobol. We just took the metric from their original papers, and therefore we used the same setting. Moreover, Section 4.2 shows that RISE and Sobol need more samples to obtain relevant explanations.
>
> ***2. In Table 2, why use 5000 samples for D-RISE but fewer for your method?***
> As above, we also took the same parameters as the D-RISE's paper.
>
> ***3. What does it mean for a method to score better on Insertion than Deletion?***
> Since Deletion measures a drop in the score, the faster the score drops, the better the metric. Hence, Deletion will favor methods that sharply identify important regions. On the contrary, since Insertion starts from an arbitrary baseline image, if the explanation map is more spread out, more relevant secondary information will be added, so the score will be better. As we can see in the maps of Appendix C, RISE saliency maps are way more spread out than HSIC's, which are sharper. It may explain why RISE is better in the Insertion benchmark and why HSIC attribution method dominates the Deletion benchmark (Section 4.1 Table 1). We provide additional quantitative examples to illustrate this link in Appendix F. Note that even if RISE dominates Insertion, it is far behind in Deletion. This is not the case for HSIC, which is still competitive in Insertion while dominating Deletion. We add those discussions in Appendix F.

---

> > ### Author Response · Authors · 2022-08-01
> > **Official Answer from Paper1305 Authors to Reviewer QhZj (2/2)**
> >
> > ## Limitations
> >
> > * In Table 1, we report the value of Insertion and Deletion metrics for HSIC method and for different grid sizes, obtained after a grid search for MobileNetV2 on 1000 ImageNet validation images. The metrics are averaged over 27 runs.
> >
> > | grid size                  | 5    | 6    | 7    | 8    | 9    | 10   |
> > | -------------------------- | ---- | ---- | ---- | ---- | ---- | ---- |
> > | Insertion $\times 10^{-1}$ | 4.14 | 4.02 | 3.90 | 3.72 | 3.54 | 3.40 |
> > | Deletion $\times 10^{-1}$  | 1.01 | 0.97 | 0.94 | 0.93 | 0.92 | 0.90 |
> > **Table 1**: Result of a grid search for MobileNetV2.
> >
> > These results show the effect of the grid size on the metric. Note that it corroborates the previous observation that a sharp explanation (high grid size) favors Deletion, and a spread-out explanation (low grid size) favors Insertion.
> >
> > * The choice for $k$ is defined by Proposition 1, which is a condition to using the decomposition property so it cannot change. The choice of $l$ may change, but RBF is largely adopted in the community when working with HSIC or RKHS.
> > * The choice of the bandwidth of $l$ could be performed in a more principled manner (e.g. by maximizing the maximum mean discrepancy, as recommended in [3]), but it would affect the efficiency of the method (one optimization for each explanation), so we chose to follow the common practice of selecting the median.
> >
> > [1] Julius Adebayo, Justin Gilmer, Michael Muelly, Ian Goodfellow, Moritz Hardt, and Been Kim. Sanity checks for saliency maps. Advances in Neural Information Processing Systems (NeurIPS), 2018
> >
> > [2] Amirata Ghorbani, Abubakar Abid, and James Zou. Interpretation of neural networks is fragile. In Proceedings of the AAAI Conference on Artificial Intelligence (AAAI), 2017
> >
> > [3] Kernel choice and classifiability for rkhs embeddings of probability distributions, Fukumizu, K., NeurIPS 2009

---

> > ### Comment · Reviewer_QhZj · 2022-08-05
> > **Update given author revisions and response**
> >
> > ### RE W1: Technical communication
> >
> > I appreciate the updated text in Sec 3.1 and Sec 3.2.
> > There's been good progress toward addressing my concerns, but there remain some lingering concerns:
> >
> > * In Sec 3.1: I think M still needs a "plain talk" definition... yes it is a binary vector in d-dim space, but what does it represent? is it just indicating over all possible patches of a certain size, which ones are masked and which are not? I think even the revised notation is trying to be too general and is going to confuse many readers, even those with lots of image processing experience.
> >
> > * In Sec. 3.2: I still find it disorienting that previously in Sec 3.1 symbol "x" denoted image pixels, but now in 3.2 "x" denotes a mask (line 159 of revised version from Aug 1). You could at least prepare the reader to make this mental switch.
> >
> > * (not that important): citing 5 papers [17 , 40 , 61, 36, 13 ] seems excessive for defining a simple inpainting operation
> >
> >
> > ## RE W2: Orthogonal decomposition property in particular is not well explained or justified
> >
> > The new example in the appendix is helpful, thanks!
> >
> > A few issues remain though, looking especially at Table 2 in App A:
> >
> > * How can HSIC be negative, if it is defined as a distance? Is this an error in approximation or something? (I am likely missing something obvious...)
> > * Suppose I decide in practice to treat any negative HSIC values as a 0 (since negative values don't "make sense"). Don't I essentially then draw the same conclusions using both RBF (where the property does NOT hold) and Sobolev?
> > * The column titles in Table 2 have some typos: (x2,y) is duplicated
> > * Shouldnt the HSIC metrics in Table 1 and Table 2 of App A match for x2, y when using the RBF kernel? Why do they differ?
> >
> > ## RE Q1-2: Fair comparisons
> >
> > OK so you use previously picked "defaults" that prefer accuracy over speed. I guess this should just be stated more clearly, perhaps indicating in the caption/table what you did for each method.
> >
> > Ideally, you could include an "efficient" version of RISE/Sobol that uses a much lower number of perturbations. Perhaps the figure in Sec 4.2 is enough to suggest this won't work very well, but that figure measures something very different than what's reported in Tab 1.
> >
> > ## RE Q3 Why deletion but not insertion
> >
> > Thanks for the detailed investigation. Please include in the final paper

---

> > > ### Author Response · Authors · 2022-08-08
> > > **Answer 2 from Paper1305 Authors to Reviewer QhZj**
> > >
> > > Thank you for your response. The current page limit will make it difficult to add more details to the revised manuscript, so we include our suggestions in this response and will add them to the paper upon acceptance.
> > > ## W1
> > > *  Line 130, in place of "For each image patch, the mask values define a drift from the original image towards the baseline value µ.," we suggest writing:  "Hence, the mask $\mathbf{M}$ aggregates the patch-wise random perturbations $M_i$ that are sampled independently for each patch ($M_i$ are iid). In practice, the perturbations contained in the mask are binary perturbations, to simulate whether the information contained in the patch is kept in the image or not."
> > > * In Sec 3.2, the $x_i$'s denote general samples of any random variable. This notation is used to introduce the general definition of $\mathcal{H}^p_{\\textnormal{x}, \\textnormal{y}}$. Indeed, each $x_i$ does not denote a mask but a patch, so we remove the confusing words "(e.g. random masks).".  In return, we further expand the added description l. 162 "... complexity. *In this work, the input variables $x_i$ are the patch perturbations $M_i$.* Therefore, we compute ... "
> > > * We keep the most cited references.
> > > ## W2
> > > * L. 604, we indeed write that "a negative value has no meaning since the metric is a distance". This negative value is obtained using the formula  $HSIC_{inter}( x_{1,2} , y) = HSIC(x_{1,2} , y) - HSIC(x_1, y) - HSIC(x_2, y)$ in a case where the decomposition property is not satisfied. HSIC is supposed to be a distance, so obtaining a negative value demonstrates that this formula does not make sense without this property.
> > > * When the decomposition property does not hold, the formula does not make sense. Therefore, even if thresholding HSIC to be positive would lead to the correct conclusion in the present example, nothing guarantees that the opposite effect, where the interactions would be overestimated, will not occur in another case.
> > > * We corrected the typos.
> > > * The metrics differ because they are not computed with the same kernel (RBF vs Sobolev). In addition, for HSIC values that are so low, the approximation error might have some impact on the estimation.
> > > ## Q1-2
> > > We will add this information to the main paper upon acceptance, thanks to the increased page limit.
> > >
> > > Empirically, the figure in Sec 4.2 shows that until a high number of samples, the explanation maps for RISE and Sobol poorly correlate with the "asymptotical baseline". This alone indicates that the HSIC explanation is converging faster to its final state (and thus its final fidelity score) whereas RISE and Sobol are oscillating and suffering from high estimation noise before converging. Theoretically, for Sobol,  Theorem 1 of [1] states that the estimation error is in $\\mathcal{O}(\frac{1}{\sqrt{p}})$ for $p  \times (d+2)$ samples, with $d$ the number of patches. In the Sobol paper [2], the number of patches considered is $121$, so for 764 samples, the estimation error would be in $\\mathcal{O}(\frac{1}{\sqrt{6}})$ (because $p  \times (d+2) = 764$ leads to $p \approx 6$ for $d = 121$). In that case, evaluating the performances of these metrics would not make sense: it could be better or worse, out of randomness.
> > >
> > > Nonetheless, we include a new study in Appendix G, showing the evolution of Deletion for HSIC, RISE and Sobol with respect to the number of samples. It shows that RISE and Sobol need more samples to reach their final deletion score.
> > > ## Q3
> > > This discussion is included in Appendix F.2. Upon acceptance, we will include it in the paper.
> > >
> > > Thank you for this constructive discussion. We believe that, as a result, our paper will be significantly improved.
> > >
> > > [1] Making best use of model evaluations to compute sensitivity indices, Saltelli, Computer Physics Communications, 2002
> > >
> > > [3] Look at the Variance! Efficient Black-box Explanations with Sobol-based Sensitivity Analysis, T. Fel, R. Cadene, M. Chalvidal, M. Cord, D. Vigouroux, and T. Serre, NeurIPS 2021

---

### Official Review · Reviewer_Zkzj · 2022-07-10

**Rating:** 7
**Confidence:** 2
**Soundness:** 4 excellent
**Presentation:** 3 good
**Contribution:** 3 good

**Summary:**

The authors present a black-box attribution method based on Hilbert-Schmidt Independence Criterion. By leveraging representation capabilities of RKHS, this method provides a dependence measure to capture more diverse information than traditional variance-based indices and does so more efficiently. This method randomly perturbs an input image in multiple regions at a time (similar to LIME, RISE, and Sobol) to identify higher-order interactions between patches. They demonstrate it on various applications including image classification and object detection tasks.  The authors quantify the efficacy of explanations using Deletion, Insertion, and uFidelity. The results show that their method is largely better than other attribution methods across a variety of neural networks, old and new.

**Questions:**

- The difference in explanations between Hi and Hixj is surprising that they are different. Shouldn't attributions from each image patch overlap with the attributions from interactions? Second-order effects are usually smaller.
- Is there an XOR situation that could highlight that Hixj is able to uncover unique interactions that would be missed by attributions of individual patches?
- Is there a way to quantify expected interactions?

**Limitations:**

The challenge is that quantitative comparisons is difficult without ground truth. The Deletion, Insertion and uFidelity metrics provide a quantitative way to compare methods but they are all flawed. Moreover, these explanations provided by these methods are quite limited as regions are highlighted. It is not clear what about those regions, i.e. edges, textures, shapes, are important. A section on limitations of attribution maps should be provided as the language term explanation used in this paper is a bit loose. Notwithstanding, it has the makings of a valuable contribution for attribution-based interpretations with clear benefits compared to existing methods.

**Strengths And Weaknesses:**

Strengths
- good job of placing research in broader field
- the theory is well founded and the implementation algorithm is clear
- the experiments and results are clear
- thorough exploration across popular attribution methods and across various architectures with better performance and faster execution times

Weakness
- The spatial interactions in the model section is anecdotal.

---

> ### Author Response · Authors · 2022-08-01
> **Official Answer from Paper1305 Authors to Reviewer Zkzj**
>
> Thank you for your positive feedback. As you noticed, finding quantitative comparison is challenging without ground truth. That is why beating the state-of-the-art in terms of Insertion and Deletion was not our only focus:
>
> * Our method is far more efficient than other SOTA black-box methods
> * Bringing the RKHS framework to the domain of AI Explainability (XAI) enables many research perspectives
> * This framework makes it possible to assess the importance of patch-wise interactions.
>
> You and the other reviewers shared concerns about the clarity of the last point, so following your suggestion, we introduce a simple pedagogical example described in an openreview general comment above **"Motivating example for orthogonal decomposition property"**. We recommend you to read it before going on with this response.
>
> **Q1:** ***Shouldn't attributions from each image patch overlap with the attributions from interactions?***
> The fact that $\mathcal{H}_i$ and $\mathcal{H}_{i\times j}$ are different is an empirical example that sometimes, interactions are as important as individual effects. The additional motivating example provided shows a simple case where it can happen. It also shows that the orthogonal decomposition property allows properly assessing interactions by removing individual effects from interaction effects, which explains why the heatmaps do not always overlap.
>
> **Q2:** ***Is there an XOR situation that could highlight that Hixj is able to uncover unique interactions that would be missed by attributions of individual patches?***
> We followed your suggestion and provided an XOR example where unique interactions would be missed by a classical attribution metric (cf **"Motivating example for orthogonal decomposition property"** in openreview general comment above)
>
> **Q3:** ***Is there a way to quantify expected interactions?***
> The orthogonal decomposition property allows to quantify these interaction effects properly and even to compare their importance with that of individual effects, as shown by the motivating example. For instance, in Section 4.4, thanks to this property, we are able to compare the importance of the interactions of the mustaches of a puma with its eye (and we find out that the interactions are more important).

---

> ### Comment · Reviewer_Zkzj · 2022-08-09
> **concerns were addressed**
>
> Thank you for addressing my concerns. This work is interesting and adds value to the field.

---

### Official Review · Reviewer_dSej · 2022-07-11

**Rating:** 5
**Confidence:** 2
**Soundness:** 3 good
**Presentation:** 3 good
**Contribution:** 2 fair

**Summary:**

Summary
The paper is motivated to improve the explainability of vision inference results. Specifically, they propose to measure the dependence between certain perturbation of regions of the input images and the actual labels in RKHSs. This helps to understand how much independent $M_i$ is from $y$, so that the approach may determine how important each perturbed fraction of the image are associated with the final inference results. The approach then is able to locate the pivotal input parts that most possible has led to the final inference. Empirical results validate that, in certain scenarios, the proposed method provides better explanation in comparison to current SOTA methods.


**Questions:**

1. How the decomposition property is related to the actual explainability of the proposed method (See details above in the weakness).
2. How the algorithm flow is connected to the eventual implementation of pixelwise explanation of the input image (See comments above regarding the algorithm flow)?

**Limitations:**

Yes, the authors have discussed the limitations.

**Strengths And Weaknesses:**

* Strength:

The method proposes to introduce the HSIC metric in order to detect the dependence between certain fraction of the input and the actual label. The approach correspondingly gives analysis in why measuring such dependence help better explain the inference result through joint interactions among  different patches, a perspective that other SOTA methods haven’t examined. Empirical results show that the proposed method can better locate the essential parts of the image.

* Weakness:

I am not quite sure how "Proposition 1", i.e., the decomposition property is related to the actual explainability of the proposed method. Is it a condition to enable the incorporation of HSIC metric in this specific scenario? It seems to me there is some gap here between the analysis and the actual motivation to improve explanation. In other words, why “it does not ensure that we assess the importance of the interactions only (line 199-200)” if the decomposition property does not hold? And how this links to the eventual inference?

The description of the algorithm could be further clarified. The current algorithm flow is a bit vague in how we can obtain the eventual explanation and to diagnose the model. Take for example, how do we link to the eventual pixel level highlights of the image parts (explanation like image Figure 3) through the construction of Eq.(2)?


* Minor issues:
1. There are some presentation issues in technical details:
Eq. (2), the brackets of the trace operation are missing.

2. There are some minor grammar issues that could be further improved. Here are some examples.
  Line 141-142 This sentence has grammar issues and needs to be rephrased;
  Line 177 Let ${x_1…X_n} $ a set of -> Let ${x_1…X_n}$ be a set of;
  Line 190 Let a Bernoulli Variable -> Let x be a Bernoulli variable;

---

> ### Author Response · Authors · 2022-08-01
> **Official Answer from Paper1305 Authors to Reviewer dSej**
>
> Thank you for your detailed feedback. We have addressed the minor issues directly in the revised version.
> ## 1. Comments on decomposition property
> ### HSIC attribution method does not require Proposition 1 to operate
> ***"Is it a condition to enable the incorporation of HSIC metric in this specific scenario?"*** Theoretically, HSIC attribution method could be used with any kernel $k$ for embedding the perturbations $M_i$. Hence, this proposition is not a condition to enable using HSIC as an attribution method. Proposition 1 only describes the conditions needed to obtain the decomposition property.
> ### Link between decomposition property and explainability
> In order to explain a model prediction, some areas of the image may only be important in interaction with other areas, affecting the output only when both areas are perturbed jointly. However, properly assessing interactions is non-trivial. The decomposition property allows to simply and correctly assess interactions. It is illustrated in an openreview general comment above, **"Motivating example for orthogonal decomposition property"**. The described experiment illustrates why Eq. 5 of the manuscript ***" does not ensure that we assess the importance of the interactions only"*** if the decomposition property does not hold. Indeed, in that case, the conditions to be able to use Eq. 5 are not fulfilled, so one cannot use it to assess the importance of the patch-wise interactions.
> ## 2. Pixel-wise aspect of explanation maps
> If we choose a grid of size $7\times 7$ , so will be the dimension of the HSIC heatmap. To be able to overlap it with original images, we need to upsample them. The pixel-wise aspect simply comes from using a bilinear upsampling, as in [1,2]. We include this comment in the revised version.
>
> [1] Look at the Variance!  Efficient Black-box Explanations with Sobol-based Sensitivity Analysis, T. Fel, R. Cadene, M. Chalvidal, M. Cord, D. Vigouroux, and T. Serre, NeurIPS 2021
>
> [2] Rise: Randomized input sampling for explanation of black-box models. V. Petsiuk, A. Das, and K. Saenko. In Proceedings of the British Machine Vision Conference (BMVC), 2018.

---

> > ### Comment · Reviewer_dSej · 2022-08-09
> > **Thanks for your response**
> >
> > Thanks for your response and further explanations. I like the clarification on the patch interaction part. I increase the score accordingly.

---

### Author Response · Authors · 2022-08-01
**Motivating example for orthogonal decomposition property**

Let us demonstrate the importance of interactions and the decomposition property with a simple example (similar to an XOR toy example, as suggested by Reviewer **Zkzj**). Let $f: [0,2]^3 \rightarrow \{0,1\}$ such that
$$
       \\textnormal{y} = f(x_1, x_2, x_3) =
   \\begin{dcases}
    1 & \text{if  } x_1 \in [0,1], x_2 \in [1,2], x_3 \in [0,1],\\\\
    1 & \text{if  } x_1 \in [0,1], x_2 \in [0,1], x_3 \in [1,2],\\\\
    0 & \text{otherwise.  }
    \\end{dcases}
$$
The function $f$ is illustrated in Appendix A. Here, $x_i$ is analogous to $M_i$. In that case, it is clear that $x_1$ is important to explain the output. However, assessing the effect of $x_2$ and $x_3$ is more difficult. They are clearly important to explain the output $\textnormal{y}$, but it can be shown theoretically that $HSIC(x_2, \textnormal{y})=0$ and $HSIC(x_3, \textnormal{y})=0$. **This motivates to assess the interactions between input variables**. One way to retrieve the information that $x_2$ and $x_3$ are important is to assess $HSIC( x_{2,3} , \textnormal{y})$, where $x_{2,3} = (x_2,x_3)$

As noticed by Reviewer **eNpV**, one could assess $HSIC( x_{2,3} , \textnormal{y})$ without any constraints on the kernel $k$, and the obtained value for $HSIC( x_{2,3} , \textnormal{y})$ would indeed be $>0$ . However, by doing so, we would also obtain that $HSIC( x_{1,2} , \textnormal{y}) > 0$ and $HSIC( x_{1,3} , \textnormal{y}) > 0$, whereas $x_1$ does not interact with $x_2$ and $x_3$, only because of the individual effect of $x_1$. We empirically illustrate this by assessing these metrics using the estimator of Eq. 2 with $p=10000$, and kernels $k$, $l$ chosen as the Radial Basis Function (RBF). The results are found in Table 1 below and show that:
* $HSIC(x_2, \textnormal{y}) \approx HSIC(x_3, \textnormal{y}) \approx 0$
* $HSIC( x_{1,2} , \textnormal{y}) \approx HSIC( x_{1,3} , \textnormal{y}) > HSIC( x_{2,3}, \textnormal{y})$

| $HSIC(x_1, \textnormal{y})$ | $HSIC(x_2, \textnormal{y})$ | $HSIC(x_2, \textnormal{y})$ | $HSIC( x_{1,3} , \textnormal{y})$ | $HSIC( x_{1,2} , \textnormal{y})$ | $HSIC( x_{2,3} , \textnormal{y})$ |
| - | - | - | - | - | - |
| $1.79 \times 10^{-2}$                    | $2.28 \times 10^{-6}$                    | $9.63 \times 10^{-6}$                    | $1.36 \times 10^{-2}$                          | $1.36 \times 10^{-2}$                          | $2.92 \times 10^{-3}$                          |
**Table 1**: HSIC metrics with $k$ taken as RBF

In order to correctly assess the pairwise interactions of input variables $x_1$ and $x_2$, one has to remove the individual effect of each variable from the $HSIC( x_{1,2} , \textnormal{y})$. The orthogonal decomposition property [1] allows to do so by simply computing $HSIC_{inter}( x_{1,2} , \textnormal{y})$ as

$HSIC_{inter}( x_{1,2} , \textnormal{y}) = HSIC( x_{1,2} , \textnormal{y}) - HSIC(x_1, \textnormal{y}) - HSIC(x_2, \textnormal{y})$

**If the decomposition property does not hold, we are not guaranteed to fully remove the individual effect of $x_1$ and $x_2$** **using the previous formula**. We estimate $HSIC_{inter}( x_{1,2} , \textnormal{y})$ when the kernel $k$ satisfies the decomposition property (in that case, we choose a Sobolev kernel as in [1]), and when it does not, and show that the correct information of  $HSIC_{inter}( x_{1,2} , \textnormal{y})$ is only retrieved when the decomposition property is satisfied. As previously, this is illustrated in the experiment, whose results are found in Table 2.

|             | $HSIC_{inter}( x_{1,2} , \textnormal{y})$ | $HSIC_{inter}( x_{1,3} , \textnormal{y})$ | $HSIC_{inter}( x_{2,3} , \textnormal{y})$ |
| ----------- | ------------------------------------------------------ | ------------------------------------------------------ | ------------------------------------------------------ |
| $k$ Sobolev | $7.68 \times 10^{-6}$                                  | $2.83 \times 10^{-6}$                                  | $7.85 \times 10^{-4}$                                  |
| $k$ RBF     | $-4.35 \times 10^{-3}$                                 | $-4.30 \times 10^{-3}$                                 | $2.91 \times 10^{-3}$                                                       |
**Table 2**: HSIC metrics for assessing interactions, when $k$ satisfies (Sobolev) / does not satisfy (RBF) the orthogonal decomposition property


In that case, with $k$ satisfying the orthogonal decomposition property (Sobolev), we retrieve that $HSIC_{inter}( x_{1,2} , \textnormal{y}) \approx HSIC_{inter}( x_{1,3} , \textnormal{y}) \approx 0$ and $HSIC_{inter}( x_{2,3} , \textnormal{y})$ is significant. When $k$ does not satisfy the property (RBF), the values are not relevant (a negative value has no meaning since the metric is a distance)

[1] Kernel-based anova decomposition and Shapley effects–application to global sensitivity analysis, Da Veiga, 2021

---

### Author Response · Authors · 2022-08-01
**General response to reviewers**

We would like to thank the reviewers for their comments and questions. They recognized the relevance and the novelty of the proposed method. The main concerns of the reviewers were about the clarity of some aspects of our work and some technical details that we address in individual responses. We modify the paper accordingly and highlight modifications in the revised manuscript in blue.

Before going on with the individual responses, we would like to comment on the orthogonal decomposition property since all reviewers shared concerns about its clarity. First, a detailed description of the decomposition property can be found in Appendix A. Second, to further clarify that point, and as suggested by Reviewer **Zkzj**, we introduce a motivating example that aims at:
* describing a case where **interactions are important to explain a model's output**,
* showing why **the orthogonal decomposition property is necessary to assess the interactions properly**.

We include this example in a second general response entitled **"Motivating example for orthogonal decomposition property"**. We also add it to Appendix A. Upon acceptance, we will include a more detailed discussion about the decomposition property and the interactions in the main document.

---

### Meta-Review · Area_Chair_an8H · 2022-08-29

**Recommendation:** Accept
**Confidence:** Certain

**Metareview:**

The paper proposes a novel black-box explanation method.  The proposed method uses HSIC to measure the dependence between randomly-masked inputs and the corresponding outputs, and identifies relevance patches.  Based on the decomposition property, the proposed method can also find interactions between patches.  Experiments quantitatively show that the proposed method outperforms (or is comparable on some evaluation measures) existing black-box methods with less computation costs.  Quantitative gains are demonstrated by finding the cause of wrong prediction in an object detection task, and interaction between patches.

Reviewers raised concerns mainly on clarity, which the authors well addressed.  I expect that the presentation of the final version will be much clearer.

A good paper with an interesting idea of using HSIC, which brings benefit on the explanation performance and computation time.  Furthermore, it allows explaining interactions, which most existing methods do not.  The advantages of the proposed methods are demonstrated quantitatively and qualitatively.

**Award:**

No

---

### Decision · Program_Chairs · 2022-09-14

Accept